# MODEL BASED REINFORCEMENT LEARNING
# FOR ATARI

**Łukasz Kaiser**[1,*]**, Mohammad Babaeizadeh**[1,*]**, Piotr Miłos**[2,3,*]**, Błażej Osiński**[2,4,*]
**Roy H. Campbell**[5]**, Konrad Czechowski**[4]**, Dumitru Erhan**[1]**, Chelsea Finn**[1,6]**,**
**Piotr Kozakowski**[4]**, Sergey Levine**[1]**, Afroz Mohiuddin**[1]**, Ryan Sepassi**[1]**,**
**George Tucker**[1]**, Henryk Michalewski**[4]

[1]Google Brain, [2]deepsense.ai, [3]Institute of Mathematics of the Polish Academy of Sciences,
[4]Faculty of Mathematics, Informatics and Mechanics, University of Warsaw,
[5]University of Illinois at Urbana–Champaign, [6]Stanford University

## ABSTRACT

Model-free reinforcement learning (RL) can be used to learn effective policies
for complex tasks, such as Atari games, even from image observations. However,
this typically requires very large amounts of interaction – substantially more, in
fact, than a human would need to learn the same games. How can people learn so
quickly? Part of the answer may be that people can learn how the game works and
predict which actions will lead to desirable outcomes. In this paper, we explore how
video prediction models can similarly enable agents to solve Atari games with fewer
interactions than model-free methods. We describe Simulated Policy Learning
(SimPLe), a complete model-based deep RL algorithm based on video prediction
models and present a comparison of several model architectures, including a novel
architecture that yields the best results in our setting. Our experiments evaluate
SimPLe on a range of Atari games in low data regime of 100k interactions between
the agent and the environment, which corresponds to two hours of real-time play.
In most games SimPLe outperforms state-of-the-art model-free algorithms, in some
games by over an order of magnitude.

## 1 INTRODUCTION

Human players can learn to play Atari games in minutes (Tsividis et al., 2017). However, some of
the best model-free reinforcement learning algorithms require tens or hundreds of millions of time
steps – the equivalent of several weeks of training in real time. How is it that humans can learn these
games so much faster? Perhaps part of the puzzle is that humans possess an intuitive understanding
of the physical processes that are represented in the game: we know that planes can fly, balls can roll,
and bullets can destroy aliens. We can therefore predict the outcomes of our actions. In this paper,
we explore how learned video models can enable learning in the Atari Learning Environment (ALE)
benchmark Bellemare et al. (2015); Machado et al. (2018) with a budget restricted to 100K time steps
– roughly to two hours of a play time.

Although prior works have proposed training predictive models for next-frame, future-frame, as well
as combined future-frame and reward predictions in Atari games (Oh et al. (2015); Chiappa et al.
(2017); Leibfried et al. (2016)), no prior work has successfully demonstrated model-based control via
predictive models that achieve competitive results with model-free RL. Indeed, in a recent survey
(Section 7.2 in Machado et al. (2018)) this was formulated as the following challenge: "*So far, there
has been no clear demonstration of successful planning with a learned model in the ALE*".

Using models of environments, or informally giving the agent ability to predict its future, has
a fundamental appeal for reinforcement learning. The spectrum of possible applications is vast,
including learning policies from the model (Watter et al., 2015; Finn et al., 2016; Finn & Levine, 2017;
Ebert et al., 2017; Hafner et al., 2019; Piergiovanni et al., 2018; Rybkin et al., 2018; Sutton & Barto,

---

*Equal contribution, authors listed in random order. BO performed the work partially during an internship at
Google Brain. Correspondence to: b.osinski@mimuw.edu.pl

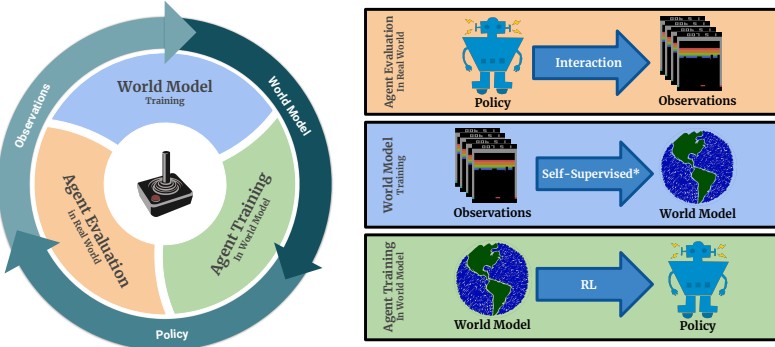

*Figure 1: Main loop of SimPLe. 1) the agent starts interacting with the real environment following the latest policy (initialized to random). 2) the collected observations will be used to train (update) the current world model. 3) the agent updates the policy by acting inside the world model. The new policy will be evaluated to measure the performance of the agent as well as collecting more data (back to 1). Note that world model training is self-supervised for the observed states and supervised for the reward.*

2017, Chapter 8), capturing important details of the scene (Ha & Schmidhuber, 2018), encouraging exploration (Oh et al., 2015), creating intrinsic motivation (Schmidhuber, 2010) or counterfactual reasoning (Buesing et al., 2019). One of the exciting benefits of model-based learning is the promise to substantially improve sample efficiency of deep reinforcement learning (see Chapter 8 in Sutton & Barto (2017)).

Our work advances the state-of-the-art in model-based reinforcement learning by introducing a system that, to our knowledge, is the first to successfully handle a variety of challenging games in the ALE benchmark. To that end, we experiment with several stochastic video prediction techniques, including a novel model based on discrete latent variables. We present an approach, called Simulated Policy Learning (SimPLe), that utilizes these video prediction techniques and trains a policy to play the game within the learned model. With several iterations of dataset aggregation, where the policy is deployed to collect more data in the original game, we learn a policy that, for many games, successfully plays the game in the real environment (see videos on the project webpage `https://goo.gl/itykP8`).

In our empirical evaluation, we find that SimPLe is significantly more sample-efficient than a highly tuned version of the state-of-the-art Rainbow algorithm (Hessel et al., 2018) on almost all games. In particular, in low data regime of 100k samples, on more than half of the games, our method achieves a score which Rainbow requires at least twice as many samples. In the best case of `Freeway`, our method is more than 10x more sample-efficient, see Figure 3. Since the publication of the first preprint of this work, it has been shown in van Hasselt et al. (2019); Kielak (2020) that Rainbow can be tuned to have better results in low data regime. The results are on a par with SimPLe – both of the model-free methods are better in 13 games, while SimPLe is better in the other 13 out of the total 26 games tested (note that in Section 4.2 van Hasselt et al. (2019) compares with the results of our first preprint, later improved).

## 2  RELATED WORK

Atari games gained prominence as a benchmark for reinforcement learning with the introduction of the Arcade Learning Environment (ALE) Bellemare et al. (2015). The combination of reinforcement learning and deep models then enabled RL algorithms to learn to play Atari games directly from images of the game screen, using variants of the DQN algorithm (Mnih et al., 2013; 2015; Hessel et al., 2018) and actor-critic algorithms (Mnih et al., 2016; Schulman et al., 2017; Babaeizadeh et al., 2017b; Wu et al., 2017; Espeholt et al., 2018). The most successful methods in this domain remain model-free algorithms (Hessel et al., 2018; Espeholt et al., 2018). Although the sample complexity of these methods has substantially improved recently, it remains far higher than the amount of experience required for human players to learn each game (Tsividis et al., 2017). In this work, we aim to learn Atari games with a budget of just 100K agent steps (400K frames), corresponding to about two hours of play time. Prior methods are generally not evaluated in this regime, and we therefore optimized Rainbow (Hessel et al., 2018) for optimal performance on 1M steps, see Appendix E for details.

Oh et al. (2015) and Chiappa et al. (2017) show that learning predictive models of Atari 2600 environments is possible using appropriately chosen deep learning architectures. Impressively, in some cases the predictions maintain low $L_2$ error over timespans of hundreds of steps. As learned simulators of Atari environments are core ingredients of our approach, in many aspects our work is motivated by Oh et al. (2015) and Chiappa et al. (2017), however we focus on using video prediction in the context of learning how to play the game well and positively verify that learned simulators can be used to train a policy useful in original environments. An important step in this direction was made by Leibfried et al. (2016), which extends the work of Oh et al. (2015) by including reward prediction, but does not use the model to learn policies that play the games. Most of these approaches, including ours, encode knowledge of the game in implicit way. Unlike this, there are works in which modeling is more explicit, for example Ersen & Sariel (2014) uses testbed of the Incredible Machines to learn objects behaviors and their interactions. Similarly Guzdial et al. (2017) learns an engine predicting interactions of predefined set of sprites in the domain of Super Mario Bros.

Perhaps surprisingly, there is virtually no work on model-based RL in video games from images. Notable exceptions are the works of Oh et al. (2017), Sodhani et al. (2019), Ha & Schmidhuber (2018), Holland et al. (2018), Leibfried et al. (2018) and Azizzadenesheli et al. (2018). Oh et al. (2017) use a model of rewards to augment model-free learning with good results on a number of Atari games. However, this method does not actually aim to model or predict future frames, and achieves clear but relatively modest gains in efficiency. Sodhani et al. (2019) proposes learning a model consistent with RNN policy which helps to train policies that are more powerful than their model-free baseline. Ha & Schmidhuber (2018) present a way to compose a variational autoencoder with a recurrent neural network into an architecture that is successfully evaluated in the VizDoom environment and on a 2D racing game. The training procedure is similar to Algorithm 1, but only one iteration of the loop is needed as the environments are simple enough to be fully explored with random exploration. Similarly, Alaniz (2018) utilizes a transition model with Monte Carlo tree search to solve a block-placing task in Minecraft. Holland et al. (2018) use a variant of Dyna (Sutton, 1991) to learn a model of the environment and generate experience for policy training in the context of Atari games. Using six Atari games as a benchmark Holland et al. (2018) measure the impact of planning shapes on performance of the Dyna-DQN algorithm and include ablations comparing scores obtained with perfect and imperfect models. Our method achieves around 330% of the Dyna-DQN score on Asterix, 120% on Q-Bert, 150% on Seaquest and 80% on Ms. Pac-Man. Azizzadenesheli et al. (2018) propose an algorithm called Generative Adversarial Tree Search (GATS) and for five Atari games train a GAN-based world model along with a Q-function. Azizzadenesheli et al. (2018) primarily discuss various failure modes of the GATS algorithm. Our method achieves around 64 times the score of GATS on Pong and 10 times on Breakout. [1]

Outside of games, model-based reinforcement learning has been investigated at length for applications such as robotics (Deisenroth et al., 2013). Though most of such works do not use image observations, several recent works have incorporated images into real-world (Finn et al., 2016; Finn & Levine, 2017; Babaeizadeh et al., 2017a; Ebert et al., 2017; Piergiovanni et al., 2018; Paxton et al., 2019; Rybkin et al., 2018; Ebert et al., 2018) and simulated (Watter et al., 2015; Hafner et al., 2019) robotic control. Our video models of Atari environments described in Section 4 are motivated by models developed in the context of robotics. Another source of inspiration are discrete autoencoders proposed by van den Oord et al. (2017) and Kaiser & Bengio (2018).

The structure of the model-based RL algorithm that we employ consists of alternating between learning a model, and then using this model to optimize a policy with model-free reinforcement learning. Variants of this basic algorithm have been proposed in a number of prior works, starting from Dyna Q Sutton (1991) to more recent methods that incorporate deep networks Heess et al. (2015); Feinberg et al. (2018); Kalweit & Boedecker (2017); Kurutach et al. (2018).

---

[1] Comparison with Dyna-DQN and GATS is based on random-normalized scores achieved at 100K interactions. Those are approximate, as the authors Dyna-DQN and GATS have not provided tabular results. Authors of Dyna-DQN also report scores on two games which we do not consider: Beam Rider and Space Invaders. For both games the reported scores are close to random scores, as are GATS scores on Asterix.

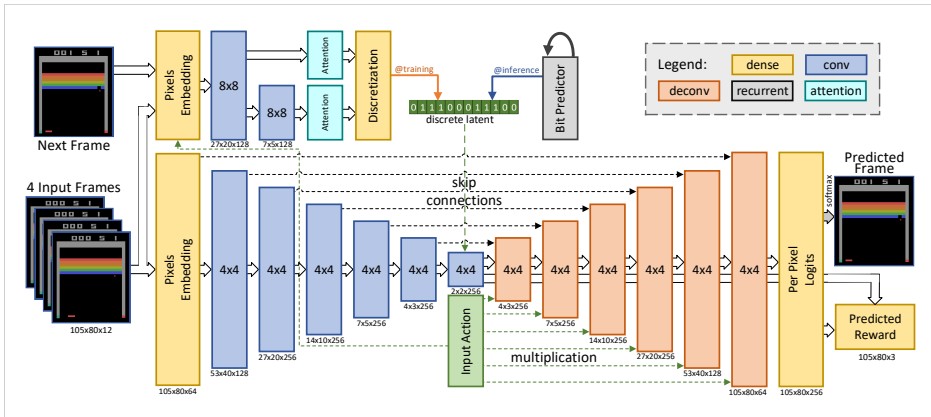

*Figure 2: Architecture of the proposed stochastic model with discrete latent. The input to the model is four stacked frames (as well as the action selected by the agent) while the output is the next predicted frame and expected reward. Input pixels and action are embedded using fully connected layers, and there is per-pixel softmax (256 colors) in the output. This model has two main components. First, the bottom part of the network which consists of a skip-connected convolutional encoder and decoder. To condition the output on the actions of the agent, the output of each layer in the decoder is multiplied with the (learned) embedded action. Second part of the model is a convolutional inference network which approximates the posterior given the next frame, similarly to Babaeizadeh et al. (2017a). At training time, the sampled latent values from the approximated posterior will be discretized into bits. To keep the model differentiable, the backpropagation bypasses the discretization following Kaiser & Bengio (2018). A third LSTM based network is trained to approximate each bit given the previous ones. At inference time, the latent bits are predicted auto-regressively using this network. The deterministic model has the same architecture as this figure but without the inference network.*

## 3  SIMULATED POLICY LEARNING (SIMPLE)

Reinforcement learning is formalized in Markov decision processes (MDP). An MDP is defined as a tuple $(\mathcal{S}, \mathcal{A}, P, r, \gamma)$, where $\mathcal{S}$ is a state space, $\mathcal{A}$ is a set of actions available to an agent, $P$ is the unknown transition kernel, $r$ is the reward function and $\gamma \in (0, 1)$ is the discount factor. In this work we refer to MDPs as environments and assume that environments do not provide direct access to the state (i.e., the RAM of Atari 2600 emulator). Instead we use visual observations, typically $210 \times 160$ RGB images. A single image does not determine the state. In order to reduce environment's partial observability, we stack four consecutive frames and use it as the observation. A reinforcement learning agent interacts with the MDP by issuing actions according to a policy. Formally, policy $\pi$ is a mapping from states to probability distributions over $\mathcal{A}$. The quality of a policy is measured by the value function $\mathbb{E}_\pi \left( \sum_{t=0}^{+\infty} \gamma^t r_{t+1} | s_0 = s \right)$, which for a starting state $s$ estimates the total discounted reward gathered by the agent.

In Atari 2600 games our goal is to find a policy which maximizes the value function from the beginning of the game. Crucially, apart from an Atari 2600 emulator environment $env$ we will use *a neural network simulated environment* $env'$ which we call a *world model* and describe in detail in Section 4. The environment $env'$ shares the action space and reward space with $env$ and produces visual observations in the same format, as it will be trained to mimic $env$. Our principal aim is to train a policy $\pi$ using a simulated environment $env'$ so that $\pi$ achieves good performance in the original environment $env$. In this training process we aim to use as few interactions with $env$ as possible. The initial data to train $env'$ comes from random rollouts of $env$. As this is unlikely to capture all aspects of $env$, we use the iterative method presented in Algorithm 1.

**Algorithm 1:** Pseudocode for SimPLe

Initialize policy $\pi$
Initialize model parameters $\theta$ of $env'$
Initialize empty set $\mathbf{D}$
**while** not done **do**
  ▷ collect observations from real env.
  $\mathbf{D} \leftarrow \mathbf{D} \cup \text{COLLECT}(env, \pi)$
  ▷ update model using collected data.
  $\theta \leftarrow \text{TRAIN\_SUPERVISED}(env', \mathbf{D})$
  ▷ update policy using world model.
  $\pi \leftarrow \text{TRAIN\_RL}(\pi, env')$
**end while**

## 4 WORLD MODELS

In search for an effective world model we experimented with various architectures, both new and modified versions of existing ones. This search resulted in a novel stochastic video prediction model (visualized in Figure 2) which achieved superior results compared to other previously proposed models. In this section, we describe the details of this architecture and the rationale behind our design decisions. In Section 6 we compare the performance of these models.

**Deterministic Model.** Our basic architecture, presented as part of Figure 2, resembles the convolutional feedforward network from Oh et al. (2015). The input $X$ consists of four consecutive game frames and an action $a$. Stacked convolution layers process the visual input. The actions are one-hot-encoded and embedded in a vector which is multiplied channel-wise with the output of the convolutional layers. The network outputs the next frame of the game and the value of the reward.

In our experiments, we varied details of the architecture above. In most cases, we use a stack of four convolutional layers with $64$ filters followed by three dense layers (the first two have $1024$ neurons). The dense layers are concatenated with $64$ dimensional vector with a learnable action embedding. Next, three deconvolutional layers of $64$ filters follow. An additional deconvolutional layer outputs an image of the original $105 \times 80$ size. The number of filters is either 3 or $3 \times 256$. In the first case, the output is a real-valued approximation of pixel's RGB value. In the second case, filters are followed by softmax producing a probability distribution on the color space. The reward is predicted by a softmax attached to the last fully connected layer. We used dropout equal to $0.2$ and layer normalization.

**Loss functions.** The visual output of our networks is either one float per pixel/channel or the categorical 256-dimensional softmax. In both cases, we used the *clipped loss* $\max(Loss, C)$ for a constant $C$. We found that clipping was crucial for improving the models (measured with the correct reward predictions per sequence metric and successful training using Algorithm 1). We conjecture that clipping substantially decreases the magnitude of gradients stemming from fine-tuning of big areas of background consequently letting the optimization process concentrate on small but important areas (e.g. the ball in Pong). In our experiments, we set $C = 10$ for $L_2$ loss on pixel values and to $C = 0.03$ for softmax loss. Note that this means that when the level of confidence about the correct pixel value exceeds $97\%$ (as $-\ln(0.97) \approx 0.03$) we get no gradients from that pixel any longer.

**Scheduled sampling.** The model $env'$ consumes its own predictions from previous steps and due to compounding errors, the model may drift out of the area of its applicability. Following Bengio et al. (2015); Venkatraman et al. (2016), we mitigate this problem by randomly replacing in training some frames of the input $X$ by the prediction from the previous step while linearly increasing the mixing probability to $100\%$ around the middle of the first iteration of the training loop.

**Stochastic Models.** A stochastic model can be used to deal with limited horizon of past observed frames as well as sprites occlusion and flickering which results to higher quality predictions. Inspired by Babaeizadeh et al. (2017a), we tried a variational autoencoder (Kingma & Welling, 2014) to model the stochasticity of the environment. In this model, an additional network receives the input frames as well as the future target frame as input and approximates the distribution of the posterior. At each timestep, a latent value $z_t$ is sampled from this distribution and passed as input to the original predictive model. At test time, the latent values are sampled from an assumed prior $\mathcal{N}(\mathbf{0}, \mathbf{I})$. To match the assumed prior and the approximate, we use the Kullback–Leibler divergence term as an additional loss term (Babaeizadeh et al., 2017a).

We noticed two major issues with the above model. First, the weight of the KL divergence loss term is game dependent, which is not practical if one wants to deal with a broad portfolio of Atari games. Second, this weight is usually a very small number in the range of $[10^{-3}, 10^{-5}]$ which means that the approximated posterior can diverge significantly from the assumed prior. This can result in previously unseen latent values at inference time that lead to poor predictions. We address these issues by utilizing a discrete latent variable similar to Kaiser & Bengio (2018).

As visualized in Figure 2, the proposed stochastic model with discrete latent variables discretizes the latent values into bits (zeros and ones) while training an auxiliary LSTM-based Hochreiter & Schmidhuber (1997) recurrent network to predict these bits autoregressively. At inference time, the latent bits will be generated by this auxiliary network in contrast to sampling from a prior. To make the predictive model more robust to unseen latent bits, we add uniform noise to approximated latent

values before discretization and apply dropout (Srivastava et al., 2014) on bits after discretization. More details about the architecture is in Appendix C.

## 5 Policy Training

We will now describe the details of SimPLe, outlined in Algorithm 1. In step 6 we use the proximal policy optimization (PPO) algorithm (Schulman et al., 2017) with $\gamma = 0.95$. The algorithm generates rollouts in the simulated environment $env'$ and uses them to improve policy $\pi$. The fundamental difficulty lays in imperfections of the model compounding over time. To mitigate this problem we use short rollouts of $env'$. Typically every $N = 50$ steps we uniformly sample the starting state from the ground-truth buffer $D$ and restart $env'$ (for experiments with the value of $\gamma$ and $N$ see Section 6.4). Using short rollouts may have a degrading effect as the PPO algorithm does not have a way to infer effects longer than the rollout length. To ease this problem, in the last step of a rollout we add to the reward the evaluation of the value function. Training with multiple iterations re-starting from trajectories gathered in the real environment is new to our knowledge. It was inspired by the classical Dyna-Q algorithm and, notably, in the Atari domain no comparable results have been achieved.

The main loop in Algorithm 1 is iterated 15 times (cf. Section 6.4). The world model is trained for 45K steps in the first iteration and for 15K steps in each of the following ones. Shorter training in later iterations does not degrade the performance because the world model after first iteration captures already part of the game dynamics and only needs to be extended to novel situations.

In each of the iterations, the agent is trained inside the latest world model using PPO. In every PPO epoch we used 16 parallel agents collecting 25, 50 or 100 steps from the simulated environment $env'$ (see Section 6.4 for ablations). The number of PPO epochs is $z \cdot 1000$, where $z$ equals to 1 in all passes except last one (where $z = 3$) and two passes number 8 and 12 (where $z = 2$). This gives 800K·$z$ interactions with the simulated environment in each of the loop passes. In the process of training the agent performs 15.2M interactions with the simulated environment $env'$.

## 6 Experiments

We evaluate SimPLe on a suite of Atari games from Atari Learning Environment (ALE) benchmark. In our experiments, the training loop is repeated for 15 iterations, with 6400 interactions with the environment collected in each iteration. We apply a standard pre-processing for Atari games: a frame skip equal to 4, that is every action is repeated 4 times. The frames are down-scaled by a factor of 2.

Because some data is collected before the first iteration of the loop, altogether $6400 \cdot 16 = 102,400$ interactions with the Atari environment are used during training. This is equivalent to $409,600$ frames from the Atari game (114 minutes at 60 FPS). At every iteration, the latest policy trained under the learned model is used to collect data in the real environment `env`. The data is also directly used to train the policy with PPO. Due to vast difference between number of training data from simulated environment and real environment (15M vs 100K) the impact of the latter on policy is negligible.

We evaluate our method on 26 games selected on the basis of being solvable with existing state-of-the-art model-free deep RL algorithms[2], which in our comparisons are Rainbow Hessel et al. (2018) and PPO Schulman et al. (2017). For Rainbow, we used the implementation from the Dopamine package and spent considerable time tuning it for sample efficiency (see Appendix E).

For visualization of all experiments see `https://goo.gl/itykP8` and for a summary see Figure 3. It can be seen that our method is more sample-efficient than a highly tuned Rainbow baseline on almost all games, requires less than half of the samples on more than half of the games and, on `Freeway`, is more than 10x more sample-efficient. Our method outperforms PPO by an even larger margin. We also compare our method with fixed score baselines (for different baselines) rather than counting how many steps are required to match our score, see Figure 4 for the results. For the

---

[2]Specifically, for the final evaluation we selected games which achieved non-random results using our method or the Rainbow algorithm using 100K interactions.

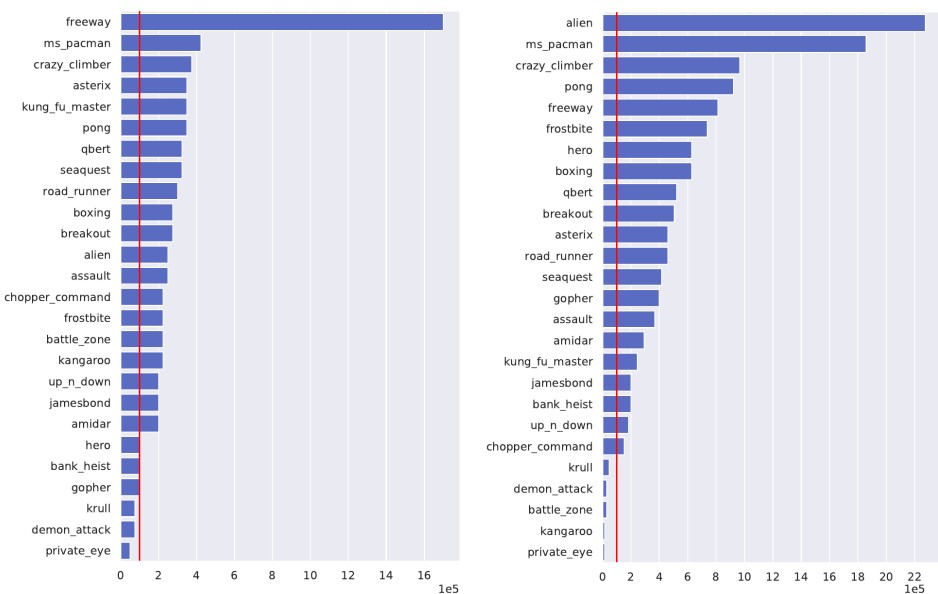

*Figure 3: Comparison with Rainbow and PPO. Each bar illustrates the number of interactions with environment required by Rainbow (left) or PPO (right) to achieve the same score as our method (SimPLe). The red line indicates the 100K interactions threshold which is used by the our method.*

qualitative analysis of performance on different games see Appendix B. The source code is available as part of the Tensor2Tensor library and it includes instructions on how to run the experiments[3].

## 6.1 SAMPLE EFFICIENCY

The primary evaluation in our experiments studies the sample efficiency of SimPLe, in comparison with state-of-the-art model-free deep RL methods in the literature. To that end, we compare with Rainbow (Hessel et al., 2018; Castro et al., 2018), which represents the state-of-the-art Q-learning method for Atari games, and PPO (Schulman et al., 2017), a model-free policy gradient algorithm (see Appendix E for details of tuning of Rainbow and PPO). The results of the comparison are presented in Figure 3. For each game, we plot the number of time steps needed for either Rainbow or PPO to reach the same score that our method reaches after 100K interaction steps. The red line indicates 100K steps: any bar larger than this indicates a game where the model-free method required more steps. SimPLe outperforms the model-free algorithms in terms of learning speed on nearly all of the games, and in the case of a few games, does so by over an order of magnitude. For some games, it reaches the same performance that our PPO implementation reaches at 10M steps. This indicates that model-based reinforcement learning provides an effective approach to learning Atari games, at a fraction of the sample complexity.

The results in these figures are generated by averaging 5 runs for each game. The model-based agent is better than a random policy for all the games except `Bank Heist`. Interestingly, we observed that the best of the 5 runs was often significantly better. For 6 of the games, it exceeds the average human score (as reported in Table 3 of Pohlen et al. (2018)). This suggests that further stabilizing SimPLe should improve its performance, indicating an important direction for future work. In some cases during training we observed high variance of the results during each step of the loop. There are a number of possible reasons, such as mutual interactions of the policy training and the supervised training or domain mismatch between the model and the real environment. We present detailed numerical results, including best scores and standard deviations, in Appendix D.

---

[3]https://github.com/tensorflow/tensor2tensor/tree/master/tensor2tensor/rl

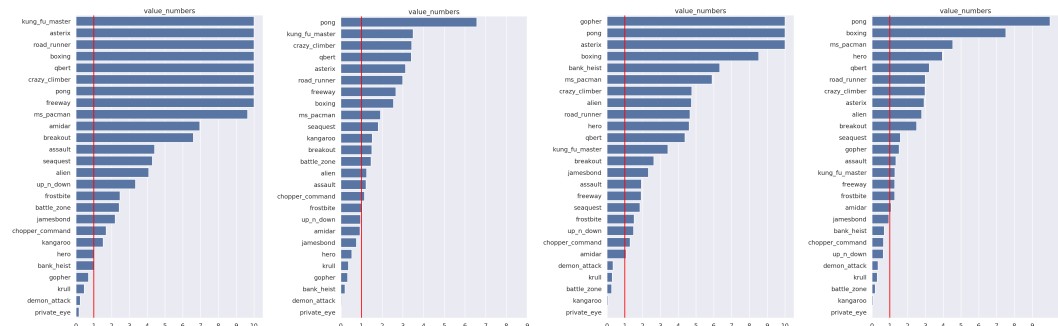

*Figure 4: Fractions of Rainbow and PPO scores at different numbers of interactions calculated with the formula* $(SimPLe\_score@100K - random\_score)/(baseline\_score - random\_score)$; *if denominator is smaller than 0, both nominator and denominator are increased by 1. From left to right, the baselines are: Rainbow at 100K, Rainbow at 200K, PPO at 100K, PPO at 200K. SimPLe outperforms Rainbow and PPO even when those are given twice as many interactions.*

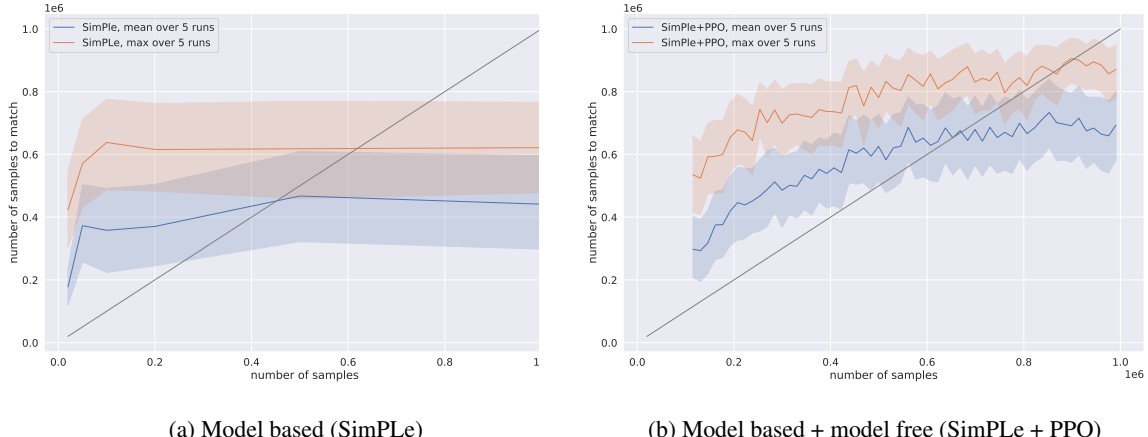

(a) Model based (SimPLe)     (b) Model based + model free (SimPLe + PPO)

*Figure 5: Behaviour with respect to the number of used samples. We report number of frames required by PPO to reach the score of our models. Results are averaged over all games.*

## 6.2 Number of frames

We focused our work on learning games with 100K interaction steps with the environment. In this section we present additional results for settings with 20K, 50K, 200K, 500K and 1M interactions; see Figure 5 (a). Our results are poor with 20K interactions. For 50K they are already almost as good as with 100K interactions. From there the results improve until 500K samples – it is also the point at which they are on par with model-free PPO. Detailed per game results can be found in Appendix F.

This demonstrates that SimPLe excels in a low data regime, but its advantage disappears with a bigger amount of data. Such a behavior, with fast growth at the beginning of training, but lower asymptotic performance is commonly observed when comparing model-based and model-free methods (Wang et al. (2019)). As observed in Section 6.4 assigning bigger computational budget helps in 100K setting. We suspect that gains would be even bigger for the settings with more samples.

Finally, we verified if a model obtained with SimPLe using 100K is a useful initialization for model-free PPO training. Based on the results depicted in Figure 5 (b) we can positively answer this conjecture. Lower asymptotic performance is probably due to worse exploration. A policy pre-trained with SimPLe was meant to obtain the best performance on 100K, at which point its entropy is very low thus hindering further PPO training.

## 6.3 Environment stochasticity

A crucial decision in the design of world models is the inclusion of stochasticity. Although Atari is known to be a deterministic environment, it is stochastic given only a limited horizon of past observed frames (in our case 4 frames). The level of stochasticity is game dependent; however, it can be observed in many Atari games. An example of such behavior can be observed in the game Kung Fu Master – after eliminating the current set of opponents, the game screen always looks the same (it contains only player's character and the background). The game dispatches diverse sets of new opponents, which cannot be inferred from the visual observation alone (without access to the game's internal state) and thus cannot be predicted by a deterministic model. Similar issues have been reported in Babaeizadeh et al. (2017a), where the output of their baseline deterministic model was a blurred superposition of possible random object movements. As can be seen in Figure 11 in the Appendix, the stochastic model learns a reasonable behavior – samples potential opponents and renders them sharply.

Given the stochasticity of the proposed model, Sim-PLe can be used with truly stochastic environments. To demonstrate this, we ran an experiment where the full pipeline (both the world model and the policy) was trained in the presence of sticky actions, as recommended in (Machado et al., 2018, Section 5). Our world model learned to account for the stickiness of actions and in most cases the end results were very similar to the ones for the deterministic case even without any tuning, see Figure 6.

## 6.4 Ablations

To evaluate the design of our method, we independently varied a number of the design decisions. Here we present an overview; see Appendix A for detailed results.

**Model architecture and hyperparameters.** We evaluated a few choices for the world model and our proposed stochastic discrete model performs best by a significant margin. The second most important parameter was the length of world model's training. We verified that a longer training would be beneficial, however we had to restrict it in all other ablation studies due to a high cost of training on all games. As for the length of rollouts from simulated $env'$, we use $N = 50$ by default. We experimentally shown that $N = 25$ performs roughly on par, while $N = 100$ is

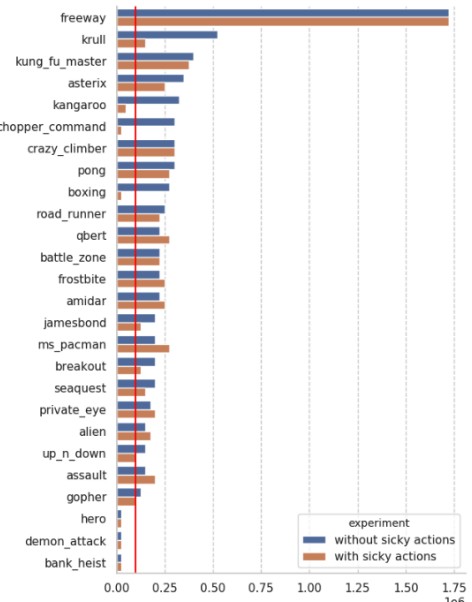

*Figure 6: Impact of the environment stochasticity. The graphs are in the same format as Figure 3: each bar illustrates the number of interactions with environment required by Rainbow to achieve the same score as SimPLe (with stochastic discrete world model) using 100k steps in an environment with and without sticky actions.*

slightly worse, likely due to compounding model errors. The *discount factor* was set to $\gamma = 0.99$ unless specified otherwise. We see that $\gamma = 0.95$ is slightly better than other values, and we hypothesize that it is due to better tolerance to model imperfections. But overall, all three values of $\gamma$ perform comparably.

**Model-based iterations.** The iterative process of training the model, training the policy, and collecting data is crucial for non-trivial tasks where random data collection is insufficient. In a game-by-game analysis, we quantified the number of games where the best results were obtained in later iterations of training. In some games, good policies could be learned very early. While this might have been due to the high variability of training, it does suggest the possibility of much faster training (i.e. in fewer step than 100k) with more directed exploration policies. In Figure 9 in the Appendix we present the cumulative distribution plot for the (first) point during learning when the maximum score for the run was achieved in the main training loop of Algorithm 1.

**Random starts.** Using short rollouts is crucial to mitigate the compounding errors in the model. To ensure exploration, SimPLe starts rollouts from randomly selected states taken from the real data buffer D. Figure 9 compares the baseline with an experiment without random starts and rollouts of length 1000 on Seaquest which shows much worse results without random starts.

# 7 CONCLUSIONS AND FUTURE WORK

We presented SimPLe, a model-based reinforcement learning approach that operates directly on raw pixel observations and learns effective policies to play games in the Atari Learning Environment. Our experiments demonstrate that SimPLe learns to play many of the games with just 100K interactions with the environment, corresponding to 2 hours of play time. In many cases, the number of samples required for prior methods to learn to reach the same reward value is several times larger.

Our predictive model has stochastic latent variables so it can be applied in highly stochastic environments. Studying such environments is an exciting direction for future work, as is the study of other ways in which the predictive neural network model could be used. Our approach uses the model as a learned simulator and directly applies model-free policy learning to acquire the policy. However, we could use the model for planning. Also, since our model is differentiable, the additional information contained in its gradients could be incorporated into the reinforcement learning process. Finally, the representation learned by the predictive model is likely be more meaningful by itself than the raw pixel observations from the environment. Incorporating this representation into the policy could further accelerate and improve the reinforcement learning process.

While SimPLe is able to learn more quickly than model-free methods, it does have limitations. First, the final scores are on the whole lower than the best state-of-the-art model-free methods. This can be improved with better dynamics models and, while generally common with model-based RL algorithms, suggests an important direction for future work. Another, less obvious limitation is that the performance of our method generally varied substantially between different runs on the same game. The complex interactions between the model, policy, and data collection were likely responsible for this. In future work, models that capture uncertainty via Bayesian parameter posteriors or ensembles (Kurutach et al., 2018; Chua et al., 2018) may improve robustness. Finally, the computational and time requirement of training inside world model are substantial (see Appendix C), which makes developing lighter models an important research direction.

In this paper our focus was to demonstrate the capability and generality of SimPLe only across a suite of Atari games, however, we believe similar methods can be applied to other environments and tasks which is one of our main directions for future work. As a long-term challenge, we believe that model-based reinforcement learning based on stochastic predictive models represents a promising and highly efficient alternative to model-free RL. Applications of such approaches to both high-fidelity simulated environments and real-world data represent an exciting direction for future work that can enable highly efficient learning of behaviors from raw sensory inputs in domains such as robotics and autonomous driving.

## ACKNOWLEDGMENTS

We thank Marc Bellemare and Pablo Castro for their help with Rainbow and Dopamine. The work of Konrad Czechowski, Piotr Kozakowski and Piotr Miłoś was supported by the Polish National Science Center grants UMO-2017/26/E/ST6/00622. The work of Henryk Michalewski was supported by the Polish National Science Center grant UMO-2018/29/B/ST6/02959. This research was supported by the PL-Grid Infrastructure. In particular, Konrad Czechowski, Piotr Kozakowski, Henryk Michalewski, Piotr Miłoś and Błażej Osiński extensively used the Prometheus supercomputer, located in the Academic Computer Center Cyfronet in the AGH University of Science and Technology in Kraków, Poland. Some of the experiments were managed using `https://neptune.ai`. We would like to thank the Neptune team for providing us access to the team version and technical support.

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

Table 1: **Summary of SimPLe ablations.** *For each game, a configuration was assigned a score being the mean over 5 experiments. The best and median scores were calculated per game. The table reports the number of games a given configuration achieved the best score or at least the median score, respectively.*

| model | best | at least median |
|---|---|---|
| deterministic | 0 | 7 |
| det. recurrent | 3 | 13 |
| SD | 8 | 16 |
| SD $\gamma = 0.9$ | 1 | 14 |
| default | 10 | 21 |
| SD 100 steps | 0 | 14 |
| SD 25 steps | 4 | 19 |

All our code is available as part of the Tensor2Tensor library and it includes instructions on how to run our experiments: `https://github.com/tensorflow/tensor2tensor/tree/master/tensor2tensor/rl`.

## A    ABLATIONS

To evaluate the design of our method, we independently varied a number of the design decisions: the choice of the model, the $\gamma$ parameter and the length of PPO rollouts. The results for 7 experimental configurations are summarized in the Table 1.

**Models.**    To assess the model choice, we evaluated the following models: deterministic, deterministic recurrent, and stochastic discrete (see Section 4). Based on Table 1 it can be seen that our proposed stochastic discrete model performs best. Figures 7a and 7b show the role of stochasticity and recurrence.

**Steps.**    See Figure 7d. As described in Section 5 every $N$ steps we reinitialize the simulated environment with ground-truth data. By default we use $N = 50$, in some experiments we set $N = 25$ or $N = 100$. It is clear from the table above and Figure 7d that $100$ is a bit worse than either $25$ or $50$, likely due to compounding model errors, but this effect is much smaller than the effect of model architecture.

**Gamma.**    See Figure 8b. We used the discount factor $\gamma = 0.99$ unless specified otherwise. We see that $\gamma = 0.95$ is slightly better than other values, and we hypothesize that it is due to better tolerance to model imperfections. But overall, all three values of $\gamma$ seem to perform comparably at the same number of steps.

**Model-based iterations.**    The iterative process of training the model, training the policy, and collecting data is crucial for non-trivial tasks where simple random data collection is insufficient. In the game-by-game analysis, we quantified the number of games where the best results were obtained in later iterations of training. In some games, good policies could be learned very early. While this might have been due simply to the high variability of training, it does suggest the possibility that much faster training – in many fewer than 100k steps – could be obtained in future work with more directed exploration policies. We leave this question to future work.

In Figure 9 we present the cumulative distribution plot for the (first) point during learning when the maximum score for the run was achieved in the main training loop of Algorithm 1.

On Figure 7c we show results for experiments in which the number samples was fixed to be 100K but the number of training loop varied. We conclude that $15$ is beneficial for training.

**Long model training**    Our best results were obtained with much 5 times longer training of the world models, see Figure 8a for comparison with shorter training. Due to our resources constraints other ablations were made with the short model training setting.

**Random starts.**  Using short rollouts is crucial to mitigate the compounding errors under the model. To ensure exploration SimPLe starts rollouts from randomly selected states taken from the real data buffer D. In Figure 9 we present a comparison with an experiment without random starts and rollouts of length 1000 on `Seaquest`. These data strongly indicate that ablating random starts substantially deteriorate results.

## B  QUALITATIVE ANALYSIS

This section provides a qualitative analysis and case studies of individual games. We emphasize that we did not adjust the method nor hyperparameters individually for each game, but we provide specific qualitative analysis to better understand the predictions from the model.[4]

**Solved games.**  The primary goal of our paper was to use model-based methods to achieve good performance within a modest budget of 100k interactions. For two games, `Pong` and `Freeway`, our method, SimPLe, was able to achieve the maximum score.

**Exploration.**  `Freeway` is a particularly interesting game. Though simple, it presents a substantial exploration challenge. The chicken, controlled by the agents, is quite slow to ascend when exploring randomly as it constantly gets bumped down by the cars (see the left video `https://goo.gl/YHbKZ6`). This makes it very unlikely to fully cross the road and obtain a non-zero reward. Nevertheless, SimPLe is able to capture such rare events, internalize them into the predictive model and then successfully learn a successful policy.

However, this good performance did not happen on every run. We conjecture the following scenario in failing cases. If at early stages the entropy of the policy decayed too rapidly the collected experience stayed limited leading to a poor world model, which was not powerful enough to support exploration (e.g. the chicken disappears when moving to high). In one of our experiments, we observed that the final policy was that the chicken moved up only to the second lane and stayed waiting to be hit by the car and so on so forth.

**Pixel-perfect games.**  In some cases (for `Pong`, `Freeway`, `Breakout`) our models were able to predict the future perfectly, down to every pixel. This property holds for rather short time intervals, we observed episodes lasting up to 50 time-steps. Extending it to long sequences would be a very exciting research direction. See videos `https://goo.gl/uyfNnW`.

**Benign errors.**  Despite the aforementioned positive examples, accurate models are difficult to acquire for some games, especially at early stages of learning. However, model-based RL should be tolerant to modest model errors. Interestingly, in some cases our models differed from the original games in a way that was harmless or only mildly harmful for policy training.

For example, in `Bowling` and `Pong`, the ball sometimes splits into two. While nonphysical, seemingly these errors did not distort much the objective of the game, see Figure 10 and also `https://goo.gl/JPi7rB`.

In `Kung Fu Master` our model's predictions deviate from the real game by spawning a different number of opponents, see Figure 11. In `Crazy Climber` we observed the bird appearing earlier in the game. These cases are probably to be attributed to the stochasticity in the model. Though not aligned with the true environment, the predicted behaviors are plausible, and the resulting policy can still play the original game.

**Failures on hard games.**  On some of the games, our models simply failed to produce useful predictions. We believe that listing such errors may be helpful in designing better training protocols and building better models. The most common failure was due to the presence of very small but highly relevant objects. For example, in `Atlantis` and `Battle Zone` bullets are so small that they tend to disappear. Interestingly, `Battle Zone` has pseudo-3D graphics, which may have added to the difficulty. See videos `https://goo.gl/uiccKU`.

---

[4]We strongly encourage the reader to watch accompanying videos `https://goo.gl/itykP8`

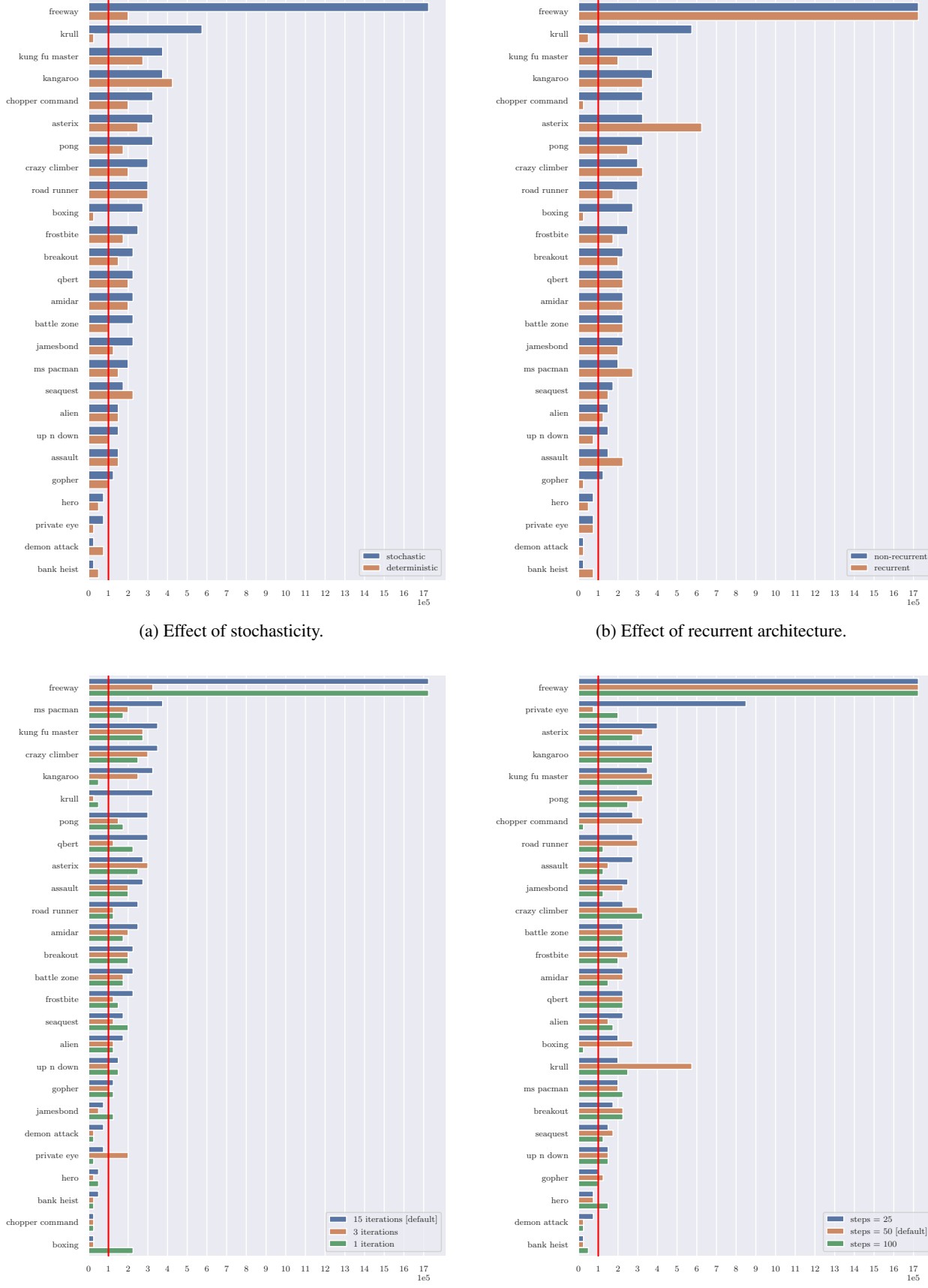

(a) Effect of stochasticity.

(b) Effect of recurrent architecture.

(c) Effect of adjusting of number of epochs.

(d) Effect of adjusting of number of steps.

Figure 7: Ablations part 1. The graphs are in the same format as Figure 3: each bar illustrates the number of interactions with environment required by Rainbow to achieve the same score as a particular variant of SimPLe. The red line indicates the 100K interactions threshold which is used by SimPLe.

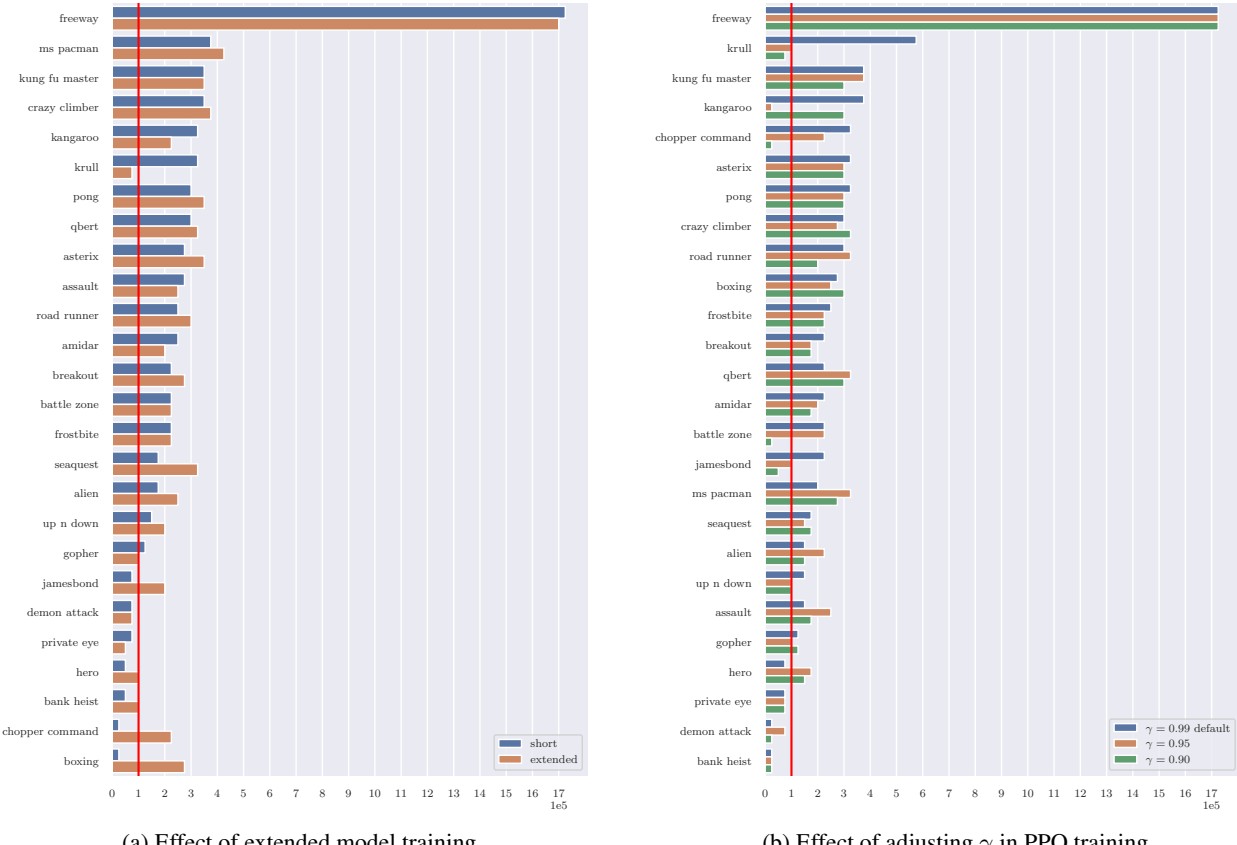

(a) Effect of extended model training.  (b) Effect of adjusting $\gamma$ in PPO training

*Figure 8: Ablations part 2. The graphs are in the same format as Figure 3: each bar illustrates the number of interactions with environment required by Rainbow to achieve the same score as a particular variant of SimPLe. The red line indicates the $100K$ interactions threshold which is used by SimPLe.*

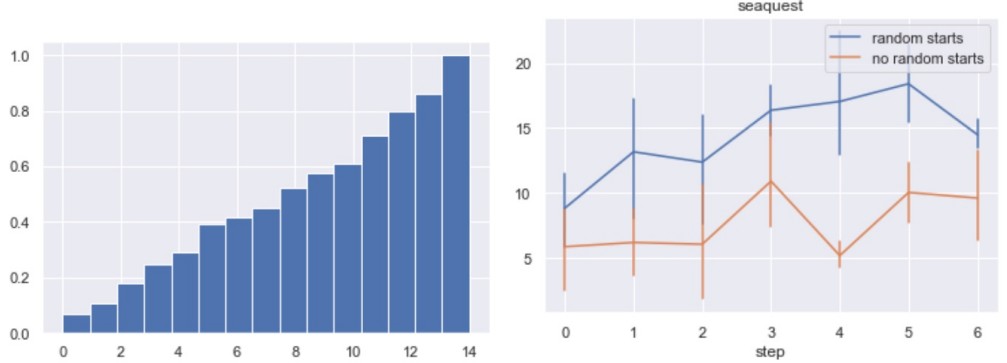

*Figure 9: (left) CDF of the number of iterations to acquire maximum score. The vertical axis represents the fraction of all games. (right) Comparison of random starts vs no random starts on* Seaquest *(for better readability we clip game rewards to $\{-1, 0, 1\}$). The vertical axis shows a mean reward and the horizontal axis the number of iterations of Algorithm 1.*

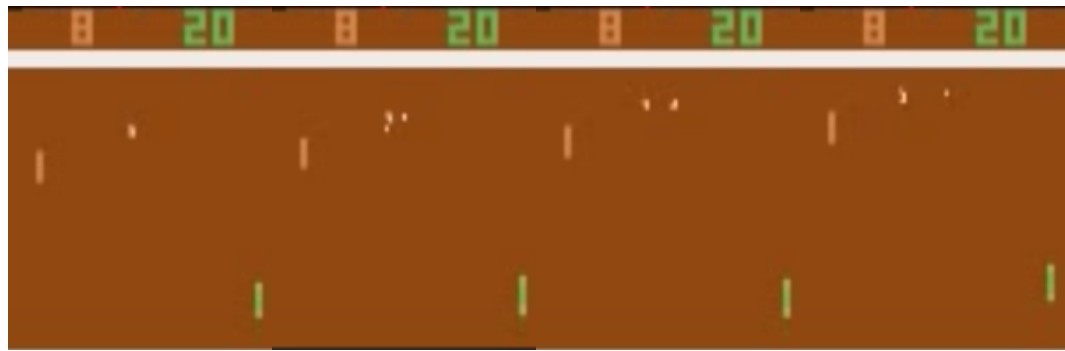

*Figure 10: Frames from the `Pong` environment.*

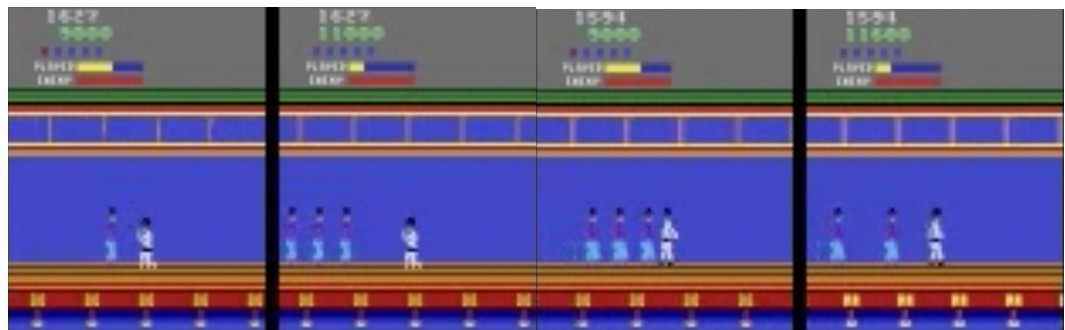

*Figure 11: Frames from the `Kung Fu Master` environment (left) and its model (right).*

Another interesting example comes from `Private Eye` in which the agent traverses different scenes, teleporting from one to the other. We found that our model generally struggled to capture such large global changes.

## C  ARCHITECTURE DETAILS

The world model is a crucial ingredient of our algorithm. Therefore the neural-network architecture of the model plays a crucial role. The high-level overview of the architecture is given in Section 4 and Figure 2. We stress that the model is general, not Atari specific, and we believe it could handle other visual prediction tasks. The whole model has around 74M parameters and the inference/backpropagation time is approx. $0.5s/0.7s$ respectively, where inference is on batch size 16 and backpropagation on batch size 2, running on NVIDIA Tesla P100. This gives us around 32ms per frame from our simulator, in comparison one step of the ALE simulator takes approximately $0.4$ms.

Below we give more details of the architecture. First, the frame prediction network:

| Layer | Number of outputs | Other details |
|---|---|---|
| Input frame dense | 96 | - |
| Downscale convolution 1 | 192 | kernel 4x4, stride 2x2 |
| Downscale convolution 2 | 384 | kernel 4x4, stride 2x2 |
| Downscale convolution 3 | 768 | kernel 4x4, stride 2x2 |
| Downscale convolution 4 | 768 | kernel 4x4, stride 2x2 |
| Downscale convolution 5 | 768 | kernel 4x4, stride 2x2 |
| Downscale convolution 6 | 768 | kernel 4x4, stride 2x2 |
| Action embedding | 768 | - |
| Latent predictor embedding | 128 | - |
| Latent predictor LSTM | 128 | - |
| Latent predictor output dense | 256 | - |
| Reward predictor hidden | 128 | - |
| Reward predictor output dense | 3 | - |
| Middle convolution 1 | 768 | kernel 3x3, stride 1x1 |
| Middle convolution 2 | 768 | kernel 3x3, stride 1x1 |
| Upscale transposed convolution 1 | 768 | kernel 4x4, stride 2x2 |
| Upscale transposed convolution 2 | 768 | kernel 4x4, stride 2x2 |
| Upscale transposed convolution 3 | 768 | kernel 4x4, stride 2x2 |
| Upscale transposed convolution 4 | 384 | kernel 4x4, stride 2x2 |
| Upscale transposed convolution 5 | 192 | kernel 4x4, stride 2x2 |
| Upscale transposed convolution 6 | 96 | kernel 4x4, stride 2x2 |
| Output frame dense | 768 | - |

The latent inference network, used just during training:

| Layer | Number of outputs | Other details |
|---|---|---|
| Downscale convolution 1 | 128 | kernel 8x8, stride 4x4 |
| Downscale convolution 2 | 512 | kernel 8x8, stride 4x4 |

All activation functions are ReLU, except for the layers marked as "output", which have softmax activations, and LSTM internal layers. In the frame prediction network, the downscale layers are connected to the corresponding upscale layers with residual connections. All convolution and transposed convolution layers are preceded by dropout 0.15 and followed by layer normalization. The latent predictor outputs 128 bits sequentially, in chunks of 8.

## D    NUMERICAL RESULTS

Below we present numerical results of our experiments. We tested SimPLe on 7 configurations (see description in Section A). For each configuration we run 5 experiments. For the evaluation of the $i$-th experiments we used the policy given by $\text{softmax}(\text{logits}(\pi_i)/T)$, where $\pi_i$ is the final learnt policy in the experiment and $T$ is the temperature parameter. We found empirically that $T = 0.5$ worked best in most cases. A tentative explanation is that polices with temperatures smaller than $1$ are less stochastic and thus more stable. However, going down to $T = 0$ proved to be detrimental in many cases as, possibly, it makes policies more prone to imperfections of models.

In Table 2 we present the mean and standard deviation of the 5 experiments. We observed that the median behaves rather similarly, which is reported it in Table 4. In this table we also show maximal scores over 5 runs. Interestingly, in many cases they turned out to be much higher. This, we hope, indicates that our methods has a further potential of reaching these higher scores.

Human scores are "Avg. Human" from Table 3 in Pohlen et al. (2018).

Table 2: Models comparison. Mean scores and standard deviations over five training runs. Right most columns presents score for random agent and human.

| Game | Ours, deterministic | Ours, det. recurrent | Ours, SD long | Ours, SD | Ours, SD $\gamma=0.90$ | Ours, SD $\gamma=0.95$ | Ours, SD 100 steps | Ours, SD 25 steps | random | human |
|---|---|---|---|---|---|---|---|---|---|---|
| Alien | 378.3 (85.5) | 321.7 (50.7) | **616.9** (252.2) | 405.2 (130.8) | 413.0 (89.7) | 590.2 (57.8) | 435.6 (78.9) | 534.8 (166.2) | 184.8 | 7128.0 |
| Amidar | 62.4 (15.2) | 86.7 (18.8) | 74.3 (28.3) | **88.0** (23.8) | 50.3 (11.7) | 78.3 (18.8) | 37.7 (15.1) | 82.2 (43.0) | 11.8 | 1720.0 |
| Assault | 361.4 (166.6) | 490.5 (143.6) | 527.2 (112.3) | 369.3 (107.8) | 406.7 (118.7) | 549.0 (127.9) | 311.7 (88.2) | **664.5** (298.2) | 233.7 | 742.0 |
| Asterix | 668.0 (294.1) | **1853.0** (391.8) | 1128.3 (211.8) | 1089.5 (335.3) | 855.0 (176.4) | 921.6 (114.2) | 777.0 (200.4) | 1340.6 (627.5) | 248.8 | 8503.0 |
| Asteroids | 743.7 (92.2) | 821.7 (115.6) | 793.6 (182.2) | 731.0 (165.3) | 882.0 (24.7) | **886.8** (45.2) | 821.9 (93.8) | 644.5 (110.6) | 649.0 | 47389.0 |
| Atlantis | 14623.4 (2122.5) | 12584.4 (5823.6) | **20992.5** (11062.0) | 14481.6 (2436.9) | 18444.1 (4616.0) | 14055.6 (6226.1) | 14139.7 (2500.9) | 11641.2 (3385.0) | 16492.0 | 29028.0 |
| BankHeist | 13.8 (2.5) | 15.1 (2.2) | **34.2** (29.2) | 8.2 (4.4) | 11.9 (2.5) | 12.0 (1.4) | 13.1 (3.2) | 12.7 (4.7) | 15.0 | 753.0 |
| BattleZone | 3306.2 (794.1) | 4665.6 (2799.4) | 4031.2 (1156.1) | **5184.4** (1347.5) | 2781.2 (661.7) | 4000.0 (788.9) | 4068.8 (2912.1) | 3746.9 (1426.8) | 2895.0 | 37188.0 |
| BeamRider | 463.8 (29.2) | 358.9 (87.4) | **621.6** (79.8) | 422.7 (103.6) | 456.2 (160.8) | 415.4 (103.4) | 456.0 (60.9) | 386.6 (264.4) | 372.1 | 16926.0 |
| Bowling | 25.3 (10.4) | 22.3 (17.0) | 30.0 (5.8) | **34.4** (16.3) | 27.7 (5.2) | 23.9 (3.3) | 29.3 (7.5) | 33.2 (15.5) | 24.2 | 161.0 |
| Boxing | -9.3 (10.9) | -3.1 (14.1) | 7.8 (10.1) | 9.1 (8.8) | **11.6** (12.6) | 5.1 (10.0) | -2.1 (5.0) | 1.6 (14.7) | 0.3 | 12.0 |
| Breakout | 6.1 (2.8) | 10.2 (5.1) | **16.4** (6.2) | 12.7 (3.8) | 7.3 (2.4) | 8.8 (5.1) | 11.4 (3.7) | 7.8 (4.1) | 0.9 | 30.0 |
| ChopperCommand | 906.9 (210.2) | 709.1 (174.1) | 979.4 (172.7) | **1246.9** (392.0) | 725.6 (204.2) | 946.6 (49.9) | 729.1 (185.1) | 1047.2 (221.6) | 671.0 | 7388.0 |
| CrazyClimber | 19380.0 (6138.8) | 54700.3 (14480.5) | **62583.6** (16856.8) | 39827.8 (22582.6) | 49840.9 (11920.9) | 34353.1 (33547.2) | 48651.2 (14903.5) | 25612.2 (14037.5) | 7339.5 | 35829.0 |
| DemonAttack | 191.9 (86.3) | 120.3 (38.3) | **208.1** (56.8) | 169.5 (41.8) | 187.5 (68.6) | 194.9 (89.6) | 170.1 (42.4) | 202.2 (134.0) | 140.0 | 1971.0 |
| FishingDerby | -94.5 (3.0) | -96.9 (1.7) | -90.7 (5.3) | -91.5 (2.8) | -91.0 (4.1) | -92.6 (3.2) | **-90.0** (2.7) | -94.5 (2.5) | -93.6 | -39.0 |
| Freeway | 5.9 (13.1) | 23.7 (13.5) | 16.7 (15.7) | 20.3 (18.5) | 18.9 (17.2) | **27.7** (13.3) | 19.1 (16.7) | 27.3 (5.8) | 0.0 | 30.0 |
| Frostbite | 196.4 (4.4) | 219.6 (21.4) | 236.9 (31.5) | **254.7** (4.9) | 234.6 (26.8) | 239.2 (19.1) | 226.8 (16.9) | 252.1 (54.4) | 74.0 | - |
| Gopher | 510.2 (158.4) | 225.2 (105.7) | 596.8 (183.5) | 771.0 (160.2) | **845.6** (230.3) | 612.6 (273.9) | 698.4 (213.9) | 509.7 (273.4) | 245.9 | 2412.0 |
| Gravitar | **237.0** (73.1) | 213.8 (57.4) | 173.4 (54.7) | 198.3 (39.9) | 213.0 (7.8) | 213.0 (37.3) | 188.9 (27.6) | 116.4 (84.0) | 227.2 | 3351.0 |
| Hero | 621.5 (1281.3) | 558.3 (1143.3) | 2656.6 (483.1) | 1295.1 (1600.1) | 2853.9 (539.5) | **3503.5** (892.9) | 3052.7 (169.3) | 1484.8 (1671.7) | 224.6 | 30826.0 |
| IceHockey | -12.6 (2.1) | -14.0 (1.8) | -11.6 (2.5) | **-10.5** (2.2) | -12.2 (2.9) | -11.9 (1.2) | -13.5 (3.0) | -13.9 (3.9) | -9.7 | 1.0 |
| Jamesbond | 68.8 (37.2) | 100.5 (69.8) | 100.5 (36.8) | 125.3 (112.5) | 28.9 (12.7) | 50.5 (21.3) | 68.9 (42.7) | **163.4** (81.8) | 29.2 | 303.0 |
| Kangaroo | **481.9** (313.2) | 191.9 (301.0) | 51.2 (17.8) | 323.1 (359.8) | 148.1 (121.5) | 37.5 (8.0) | 301.2 (593.4) | 340.0 (470.4) | 42.0 | 3035.0 |
| Krull | 834.9 (166.3) | 1778.5 (906.9) | 2204.8 (776.5) | **4539.9** (2470.4) | 2396.5 (962.0) | 2620.9 (856.2) | 3559.0 (1896.7) | 3320.6 (2410.1) | 1543.3 | 2666.0 |
| KungFuMaster | 10340.9 (8835.7) | 4086.6 (3384.5) | 14862.5 (4031.6) | **17257.2** (5502.6) | 12587.8 (6810.0) | 16926.6 (6598.3) | 17121.2 (7211.6) | 15541.2 (5086.1) | 616.5 | 22736.0 |
| MsPacman | 560.6 (172.2) | 1098.1 (450.9) | **1480.0** (288.2) | 762.8 (331.5) | 1197.1 (544.6) | 1273.3 (59.5) | 921.0 (306.0) | 805.8 (261.1) | 235.2 | 6952.0 |
| NameThisGame | 1512.1 (408.3) | 2007.9 (367.0) | **2420.7** (289.4) | 1990.4 (284.7) | 2058.1 (103.7) | 2114.8 (387.4) | 2067.2 (304.8) | 1805.3 (453.4) | 2136.8 | 8049.0 |
| Pong | -17.4 (5.2) | -11.6 (15.9) | **12.8** (17.2) | 5.2 (9.7) | -2.9 (7.3) | -2.5 (15.4) | -13.9 (7.7) | -1.0 (14.9) | -20.4 | 15.0 |
| PrivateEye | 16.4 (46.7) | 50.8 (43.2) | 35.0 (60.2) | 58.3 (45.4) | 54.4 (49.0) | 67.8 (26.4) | 88.3 (19.0) | **1334.3** (1794.5) | 26.6 | 69571.0 |
| Qbert | 480.4 (158.8) | 603.7 (150.3) | **1288.8** (1677.9) | 559.8 (183.8) | 899.3 (474.3) | 1120.2 (697.1) | 534.4 (162.5) | 603.4 (138.2) | 166.1 | 13455.0 |
| Riverraid | 1285.6 (604.6) | 1740.7 (458.1) | 1957.8 (758.1) | 1587.0 (818.0) | 1977.4 (359.8) | **2115.1** (106.2) | 1318.7 (540.4) | 1426.0 (374.0) | 1451.0 | 17118.0 |
| RoadRunner | 5724.4 (3093.1) | 1228.8 (1025.9) | 5640.6 (3936.6) | 5169.4 (3939.0) | 1586.2 (1574.1) | **8414.1** (4542.8) | 722.2 (627.2) | 4366.2 (3867.8) | 0.0 | 7845.0 |
| Seaquest | 419.5 (236.2) | 289.6 (110.4) | **683.3** (171.2) | 370.9 (128.2) | 364.6 (138.6) | 337.8 (79.0) | 247.8 (72.4) | 350.0 (136.8) | 61.1 | 42055.0 |
| UpNDown | 1329.3 (495.3) | 926.7 (335.7) | **3350.3** (3540.0) | 2152.6 (1192.4) | 1291.2 (324.6) | 1250.6 (493.0) | 1828.4 (688.3) | 2136.5 (2095.0) | 488.4 | 11693.0 |
| YarsRevenge | 3014.9 (397.4) | 3291.4 (1097.3) | **5664.3** (1870.5) | 2980.2 (778.6) | 2934.2 (459.2) | 3366.6 (493.0) | 2673.7 (216.8) | 4666.1 (1889.4) | 3121.2 | 54577.0 |

Table 3: *Comparison of our method (SimPLe) with model-free benchmarks - PPO and Rainbow, trained with 100 thousands/500 thousands/1 million steps. (1 step equals 4 frames)*

| Game | SimPLe | | PPO_100k | | PPO_500k | | PPO_1m | | Rainbow_100k | | Rainbow_500k | | Rainbow_1m | | random | human |
|---|---|---|---|---|---|---|---|---|---|---|---|---|---|---|---|---|
| Alien | 616.9 | (252.2) | 291.0 | (40.3) | 269.0 | (203.4) | 362.0 | (102.0) | 290.6 | (14.8) | 828.6 | (54.2) | 945.0 | (85.0) | 184.8 | 7128.0 |
| Amidar | 74.3 | (28.3) | 56.5 | (20.8) | 93.2 | (36.7) | 123.8 | (19.7) | 20.8 | (2.3) | 194.0 | (34.9) | 275.8 | (66.7) | 11.8 | 1720.0 |
| Assault | 527.2 | (112.3) | 424.2 | (55.8) | 552.3 | (110.4) | 1134.4 | (798.8) | 300.3 | (14.6) | 1041.5 | (92.1) | 1581.8 | (207.8) | 233.7 | 742.0 |
| Asterix | 1128.3 | (211.8) | 385.0 | (104.4) | 1085.0 | (354.8) | 2185.0 | (931.6) | 285.7 | (9.3) | 1702.7 | (162.8) | 2151.6 | (202.6) | 248.8 | 8503.0 |
| Asteroids | 793.6 | (182.2) | 1134.0 | (326.9) | 1053.0 | (433.3) | 1251.0 | (377.9) | 912.3 | (62.7) | 895.9 | (82.0) | 1071.5 | (91.7) | 649.0 | 47389.0 |
| Atlantis | 20992.5 | (11062.0) | 34316.7 | (5703.8) | 4836416.7 | (6218247.3) | - | (-) | 17881.8 | (617.6) | 79541.0 | (25393.4) | 848800.0 | (37533.1) | 16492.0 | 29028.0 |
| BankHeist | 34.2 | (29.2) | 16.0 | (12.4) | 641.0 | (352.8) | 856.0 | (376.7) | 34.5 | (2.0) | 727.3 | (198.3) | 1053.3 | (22.9) | 15.0 | 753.0 |
| BattleZone | 4031.2 | (1156.1) | 5300.0 | (3655.1) | 14400.0 | (6476.1) | 19000.0 | (4571.7) | 3363.5 | (523.8) | 19507.1 | (3193.3) | 22391.4 | (7708.9) | 2895.0 | 37188.0 |
| BeamRider | 621.6 | (79.8) | 563.6 | (189.4) | 497.6 | (103.5) | 684.0 | (168.8) | 365.6 | (29.8) | 5890.0 | (525.6) | 6945.3 | (1390.8) | 372.1 | 16926.0 |
| Bowling | 30.0 | (5.8) | 17.7 | (11.2) | 28.5 | (3.4) | 35.8 | (6.2) | 24.7 | (0.8) | 31.0 | (1.9) | 30.6 | (6.2) | 24.2 | 161.0 |
| Boxing | 7.8 | (10.1) | -3.9 | (6.4) | 3.5 | (3.5) | 19.6 | (20.9) | 0.9 | (1.7) | 58.2 | (16.5) | 80.3 | (5.6) | 0.3 | 12.0 |
| Breakout | 16.4 | (6.2) | 5.9 | (3.3) | 66.1 | (114.3) | 128.0 | (153.3) | 3.3 | (0.1) | 26.7 | (2.4) | 38.7 | (3.4) | 0.9 | 30.0 |
| ChopperCommand | 979.4 | (172.7) | 730.0 | (199.0) | 860.0 | (285.3) | 970.0 | (201.5) | 776.6 | (59.0) | 1765.2 | (280.7) | 2474.0 | (504.5) | 671.0 | 7388.0 |
| CrazyClimber | 62583.6 | (16856.8) | 18400.0 | (5275.1) | 33420.0 | (3628.3) | 58000.0 | (16994.6) | 12558.3 | (674.6) | 75655.1 | (9439.6) | 97088.1 | (9975.4) | 7339.5 | 35829.0 |
| DemonAttack | 208.1 | (56.8) | 192.5 | (83.1) | 216.5 | (96.2) | 241.0 | (135.0) | 431.6 | (79.5) | 3642.1 | (478.2) | 5478.6 | (297.9) | 140.0 | 1971.0 |
| FishingDerby | -90.7 | (5.3) | -95.6 | (4.3) | -87.2 | (5.3) | -88.8 | (4.0) | -91.1 | (2.1) | -66.7 | (6.0) | -23.2 | (22.3) | -93.6 | -39.0 |
| Freeway | 16.7 | (15.7) | 8.0 | (9.8) | 14.0 | (11.5) | 20.8 | (11.1) | 0.1 | (0.1) | 12.6 | (15.4) | 13.0 | (15.9) | 0.0 | 30.0 |
| Frostbite | 236.9 | (31.5) | 174.0 | (40.7) | 214.0 | (10.2) | 229.0 | (20.6) | 140.1 | (2.7) | 1386.1 | (321.7) | 2972.3 | (284.9) | 74.0 | - |
| Gopher | 596.8 | (183.5) | 246.0 | (103.3) | 560.0 | (118.8) | 696.0 | (279.3) | 748.3 | (105.4) | 1640.5 | (105.6) | 1905.0 | (211.1) | 245.9 | 2412.0 |
| Gravitar | 173.4 | (54.7) | 235.0 | (197.2) | 235.0 | (134.7) | 325.0 | (85.1) | 231.4 | (50.7) | 214.9 | (27.6) | 260.0 | (22.3) | 227.2 | 3351.0 |
| Hero | 2656.6 | (483.1) | 569.0 | (1100.9) | 1824.0 | (1461.2) | 3719.0 | (1306.0) | 2676.3 | (93.7) | 10664.3 | (1060.5) | 13295.5 | (261.2) | 224.6 | 30826.0 |
| IceHockey | -11.6 | (2.5) | -10.0 | (2.1) | -6.6 | (1.6) | -5.3 | (1.7) | -9.5 | (0.8) | -9.7 | (0.8) | -6.5 | (0.5) | -9.7 | 1.0 |
| Jamesbond | 100.5 | (36.8) | 65.0 | (46.4) | 255.0 | (101.7) | 310.0 | (129.0) | 61.7 | (8.8) | 429.7 | (27.9) | 692.6 | (316.2) | 29.2 | 303.0 |
| Kangaroo | 51.2 | (17.8) | 140.0 | (102.0) | 340.0 | (407.9) | 840.0 | (806.5) | 38.7 | (9.3) | 970.9 | (501.9) | 4084.6 | (1954.1) | 42.0 | 3035.0 |
| Krull | 2204.8 | (776.5) | 3750.4 | (3071.9) | 3056.1 | (1155.5) | 5061.8 | (1333.4) | 2978.8 | (148.4) | 4139.4 | (336.2) | 4971.1 | (360.3) | 1543.3 | 2666.0 |
| KungFuMaster | 14862.5 | (4031.6) | 4820.0 | (983.2) | 17370.0 | (10707.6) | 13780.0 | (3971.6) | 1019.4 | (149.6) | 19346.1 | (3274.4) | 21258.6 | (3210.2) | 616.5 | 22736.0 |
| MsPacman | 1480.0 | (288.2) | 496.0 | (379.8) | 306.0 | (70.2) | 594.0 | (247.9) | 364.3 | (20.4) | 1558.0 | (248.9) | 1881.4 | (112.0) | 235.2 | 6952.0 |
| NameThisGame | 2420.7 | (289.4) | 2225.0 | (423.7) | 2106.0 | (898.8) | 2311.0 | (547.6) | 2368.2 | (318.3) | 4886.5 | (583.1) | 4454.2 | (338.3) | 2136.8 | 8049.0 |
| Pong | 12.8 | (17.2) | -20.5 | (0.6) | -8.6 | (14.9) | 14.7 | (5.1) | -19.5 | (0.2) | 19.9 | (0.4) | 20.6 | (0.2) | -20.4 | 15.0 |
| PrivateEye | 35.0 | (60.2) | 10.0 | (20.0) | 20.0 | (40.0) | 20.0 | (40.0) | 42.1 | (53.8) | -6.2 | (89.8) | 2336.7 | (4732.6) | 26.6 | 69571.0 |
| Qbert | 1288.8 | (1677.9) | 362.5 | (117.8) | 757.5 | (78.9) | 2675.0 | (1701.1) | 235.6 | (12.9) | 4241.7 | (193.1) | 8885.2 | (1690.9) | 166.1 | 13455.0 |
| Riverraid | 1957.8 | (758.1) | 1398.0 | (513.8) | 2865.0 | (327.1) | 2887.0 | (807.0) | 1904.2 | (44.2) | 5068.6 | (292.6) | 7018.9 | (334.2) | 1451.0 | 17118.0 |
| RoadRunner | 5640.6 | (3936.6) | 1430.0 | (760.0) | 5750.0 | (5259.9) | 8930.0 | (4304.0) | 524.1 | (147.5) | 18415.4 | (5280.0) | 31379.7 | (3225.8) | 0.0 | 7845.0 |
| Seaquest | 683.3 | (171.2) | 370.0 | (103.3) | 692.0 | (48.3) | 882.0 | (122.7) | 206.3 | (17.1) | 1558.7 | (221.2) | 3279.9 | (683.9) | 61.1 | 42055.0 |
| UpNDown | 3350.3 | (3540.0) | 2874.0 | (1105.8) | 12126.0 | (1389.5) | 13777.0 | (6766.3) | 1346.3 | (95.1) | 6120.7 | (356.8) | 8010.9 | (907.0) | 488.4 | 11693.0 |
| YarsRevenge | 5664.3 | (1870.5) | 5182.0 | (1209.3) | 8064.8 | (2859.8) | 9495.0 | (2638.3) | 3649.0 | (168.6) | 7005.7 | (394.2) | 8225.1 | (957.9) | 3121.2 | 54577.0 |

Table 4: Models comparison. Scores of median (left) and best (right) models out of five training runs. Right most columns presents score for random agent and human.

| Game | Ours, deterministic | | Ours, det. recurrent | | Ours, SD long | | Ours, SD | | Ours, SD $\gamma = 0.90$ | | Ours, SD $\gamma = 0.95$ | | SD 100 steps | | Ours, SD 25 steps | | random | human |
|---|---|---|---|---|---|---|---|---|---|---|---|---|---|---|---|---|---|---|
| Alien | 354.4 | 516.6 | 299.2 | 381.1 | 515.9 | 1030.5 | 409.2 | 586.9 | 411.9 | 530.5 | 567.3 | 682.7 | 399.5 | 522.3 | 525.5 | 792.8 | 184.8 | 7128.0 |
| Amidar | 58.0 | 84.8 | 82.7 | 118.4 | 80.2 | 102.7 | 85.1 | 114.0 | 55.1 | 58.9 | 84.3 | 101.4 | 45.2 | 47.5 | 93.1 | 137.7 | 11.8 | 1720.0 |
| Assault | 334.4 | 560.1 | 566.6 | 627.2 | 509.1 | 671.1 | 355.7 | 527.9 | 369.1 | 614.4 | 508.4 | 722.5 | 322.9 | 391.1 | 701.4 | 1060.3 | 233.7 | 742.0 |
| Asterix | 529.7 | 1087.5 | 1798.4 | 2282.0 | 1065.6 | 1485.2 | 1158.6 | 1393.8 | 805.5 | 1159.4 | 923.4 | 1034.4 | 813.3 | 1000.0 | 1128.1 | 2313.3 | 248.8 | 8503.0 |
| Asteroids | 727.3 | 854.7 | 827.7 | 919.8 | 899.7 | 955.6 | 671.2 | 962.0 | 885.5 | 909.1 | 886.1 | 949.5 | 813.8 | 962.2 | 657.5 | 752.7 | 649.0 | 47389.0 |
| Atlantis | 15587.5 | 16545.3 | 15939.1 | 17778.1 | 13695.3 | 34890.6 | 13645.3 | 18396.9 | 19367.2 | 23046.9 | 12981.2 | 23579.7 | 15020.3 | 16790.6 | 12196.9 | 15728.1 | 16492.0 | 29028.0 |
| BankHeist | 14.4 | 16.2 | 14.7 | 18.8 | 31.9 | 77.5 | 8.9 | 13.9 | 12.3 | 14.5 | 12.3 | 13.1 | 12.8 | 17.2 | 14.1 | 17.0 | 15.0 | 753.0 |
| BattleZone | 3312.5 | 4140.6 | 4515.6 | 9312.5 | 3484.4 | 5359.4 | 5390.6 | 7093.8 | 2937.5 | 3343.8 | 4421.9 | 4703.1 | 3500.0 | 8906.2 | 3859.4 | 5734.4 | 2895.0 | 37188.0 |
| BeamRider | 453.1 | 515.5 | 351.4 | 470.2 | 580.2 | 728.8 | 433.9 | 512.6 | 393.5 | 682.8 | 446.6 | 519.2 | 447.1 | 544.6 | 385.7 | 741.9 | 372.1 | 16926.0 |
| Bowling | 27.0 | 36.2 | 28.4 | 43.7 | 28.0 | 39.6 | 24.9 | 55.0 | 27.7 | 34.9 | 22.6 | 28.6 | 28.4 | 39.9 | 37.0 | 54.7 | 24.2 | 161.0 |
| Boxing | -7.1 | 0.2 | 3.5 | 5.0 | 9.4 | 21.0 | 8.3 | 21.5 | 6.4 | 31.5 | 2.5 | 15.0 | -0.7 | 2.2 | -0.9 | 20.8 | 0.3 | 12.0 |
| Breakout | 5.5 | 9.8 | 12.5 | 13.9 | 16.0 | 22.8 | 11.0 | 19.5 | 7.4 | 10.4 | 10.2 | 14.1 | 10.5 | 16.7 | 6.9 | 13.0 | 0.9 | 30.0 |
| ChopperCommand | 942.2 | 1167.2 | 748.4 | 957.8 | 909.4 | 1279.7 | 1139.1 | 1909.4 | 682.8 | 1045.3 | 954.7 | 1010.9 | 751.6 | 989.1 | 1031.2 | 1329.7 | 671.0 | 7388.0 |
| CrazyClimber | 20754.7 | 23831.2 | 49854.7 | 80156.2 | 55795.3 | 87593.8 | 41396.9 | 67250.0 | 56875.0 | 58979.7 | 19448.4 | 84070.3 | 53406.2 | 64196.9 | 19345.3 | 43179.7 | 7339.5 | 35829.0 |
| DemonAttack | 219.2 | 263.0 | 135.8 | 148.4 | 191.2 | 288.9 | 182.4 | 223.9 | 160.3 | 293.8 | 204.1 | 312.8 | 164.4 | 222.6 | 187.5 | 424.8 | 140.0 | 1971.0 |
| FishingDerby | -94.3 | -90.2 | -97.3 | -94.2 | -91.8 | -84.3 | -91.6 | -88.6 | -90.0 | -85.7 | -92.0 | -88.8 | -90.6 | -85.4 | -95.0 | -90.7 | -93.6 | -39.0 |
| Freeway | 0.0 | 29.3 | 29.3 | 32.2 | 21.5 | 32.0 | 33.5 | 34.0 | 31.1 | 32.0 | 33.5 | 33.8 | 30.0 | 32.3 | 29.9 | 33.5 | 0.0 | 30.0 |
| Frostbite | 194.5 | 203.9 | 213.4 | 256.2 | 248.8 | 266.9 | 253.1 | 262.8 | 246.7 | 261.7 | 250.0 | 255.9 | 215.8 | 247.7 | 249.4 | 337.5 | 74.0 | - |
| Gopher | 514.7 | 740.6 | 270.3 | 320.9 | 525.3 | 845.6 | 856.9 | 934.4 | 874.1 | 1167.2 | 604.1 | 1001.6 | 726.9 | 891.6 | 526.2 | 845.0 | 245.9 | 2412.0 |
| Gravitar | 232.8 | 310.2 | 219.5 | 300.0 | 156.2 | 233.6 | 202.3 | 252.3 | 223.4 | 225.8 | 228.1 | 243.8 | 193.8 | 218.0 | 93.0 | 240.6 | 227.2 | 3351.0 |
| Hero | 71.5 | 2913.0 | 75.0 | 2601.5 | 2935.0 | 3061.6 | 237.5 | 3133.8 | 3135.0 | 3147.5 | 3066.2 | 5092.0 | 3067.3 | 3256.9 | 1487.2 | 2964.8 | 224.6 | 30826.0 |
| IceHockey | -12.4 | -9.9 | -14.8 | -11.8 | -12.3 | -7.2 | -10.0 | -7.7 | -11.8 | -8.5 | -11.6 | -10.7 | -12.9 | -10.0 | -12.2 | -11.0 | -9.7 | 1.0 |
| Jamesbond | 64.8 | 128.9 | 64.8 | 219.5 | 110.9 | 141.4 | 87.5 | 323.4 | 25.0 | 46.9 | 58.6 | 69.5 | 61.7 | 139.1 | 139.8 | 261.7 | 29.2 | 303.0 |
| Kangaroo | 500.0 | 828.1 | 68.8 | 728.1 | 62.5 | 65.6 | 215.6 | 909.4 | 103.1 | 334.4 | 34.4 | 50.0 | 43.8 | 1362.5 | 56.2 | 1128.1 | 42.0 | 3035.0 |
| Krull | 852.2 | 1014.3 | 1783.6 | 2943.6 | 1933.7 | 3317.5 | 4264.3 | 7163.2 | 1874.8 | 3554.5 | 2254.0 | 3827.1 | 3142.8 | 6315.2 | 3198.2 | 6833.4 | 1543.3 | 2666.0 |
| KungFuMaster | 7575.0 | 20450.0 | 4848.4 | 8065.6 | 14318.8 | 21054.7 | 17448.4 | 21943.8 | 12964.1 | 21956.2 | 20195.3 | 23690.6 | 19718.8 | 25375.0 | 18025.0 | 20365.6 | 616.5 | 22736.0 |
| MsPacman | 557.3 | 818.0 | 1178.8 | 1685.9 | 1525.0 | 1903.4 | 751.2 | 1146.1 | 1410.5 | 1538.9 | 1277.3 | 1354.5 | 866.2 | 1401.9 | 777.2 | 1227.8 | 235.2 | 6952.0 |
| NameThisGame | 1468.1 | 1992.7 | 1826.7 | 2614.5 | 2460.0 | 2782.8 | 1919.8 | 2377.7 | 2087.3 | 2155.2 | 1994.8 | 2570.3 | 2153.4 | 2471.9 | 1964.2 | 2314.8 | 2136.8 | 8049.0 |
| Pong | -19.6 | -8.5 | -17.3 | 16.7 | 20.7 | 21.0 | 1.4 | 21.0 | -2.0 | 6.6 | 3.8 | 14.2 | -17.9 | -2.0 | -10.1 | 21.0 | -20.4 | 15.0 |
| PrivateEye | 0.0 | 98.9 | 75.0 | 82.8 | 0.0 | 100.0 | 76.6 | 100.0 | 75.0 | 96.9 | 60.9 | 100.0 | 96.9 | 99.3 | 100.0 | 4038.7 | 26.6 | 69571.0 |
| Qbert | 476.6 | 702.7 | 555.9 | 869.9 | 656.2 | 4259.0 | 508.6 | 802.7 | 802.3 | 1721.9 | 974.6 | 2322.3 | 475.0 | 812.5 | 668.8 | 747.3 | 166.1 | 13455.0 |
| Riverraid | 1416.1 | 1929.4 | 1784.4 | 2274.5 | 2360.0 | 2659.8 | 1799.4 | 2158.4 | 2053.8 | 2307.5 | 2143.6 | 2221.2 | 1387.8 | 1759.8 | 1345.5 | 1923.4 | 1451.0 | 17118.0 |
| RoadRunner | 5901.6 | 8484.4 | 781.2 | 2857.8 | 5906.2 | 11176.6 | 2804.7 | 10676.6 | 1620.3 | 4104.7 | 7032.8 | 14978.1 | 857.8 | 1342.2 | 2717.2 | 8560.9 | 0.0 | 7845.0 |
| Seaquest | 414.4 | 768.1 | 236.9 | 470.6 | 711.6 | 854.1 | 386.9 | 497.2 | 330.9 | 551.2 | 332.8 | 460.9 | 274.1 | 317.2 | 366.9 | 527.2 | 61.1 | 42055.0 |
| UpNDown | 1195.9 | 2071.1 | 1007.5 | 1315.2 | 1616.1 | 8614.5 | 2389.5 | 3798.3 | 1433.3 | 1622.0 | 1248.6 | 1999.4 | 1670.3 | 2728.0 | 1825.2 | 5193.1 | 488.4 | 11693.0 |
| YarsRevenge | 3047.0 | 3380.5 | 3416.3 | 4230.8 | 6580.2 | 7547.4 | 2435.5 | 3914.1 | 2955.9 | 3314.5 | 3434.8 | 3896.3 | 2745.3 | 2848.1 | 4276.3 | 6673.1 | 3121.2 | 54577.0 |

## E    Baselines optimization

To assess the performance of SimPle we compare it with model-free algorithms. To make this comparison more reliable we tuned Rainbow in the low data regime. To this end we run an hyperparameter search over the following parameters from `https://github.com/google/dopamine/blob/master/dopamine/agents/rainbow/rainbow_agent.py`:

- `update_horizon` in {1, 3}, best parameter = 3
- `min_replay_history` in {500, 5000, 20000}, best parameter = 20000
- `update_period` in {1, 4}, best parameter = 4
- `target_update_period` {50, 100, 1000, 4000}, best parameter = 8000
- `replay_scheme` in {`uniform`, `prioritized`}, best parameter = `prioritized`

Each set of hyperparameters was used to train 5 Rainbow agents on the game of `Pong` until 1 million of interactions with the environment. Their average performance was used to pick the best hyperparameter set.

For PPO we used the standard set of hyperparameters from `https://github.com/openai/baselines`.

## F    Results at different numbers of interactions

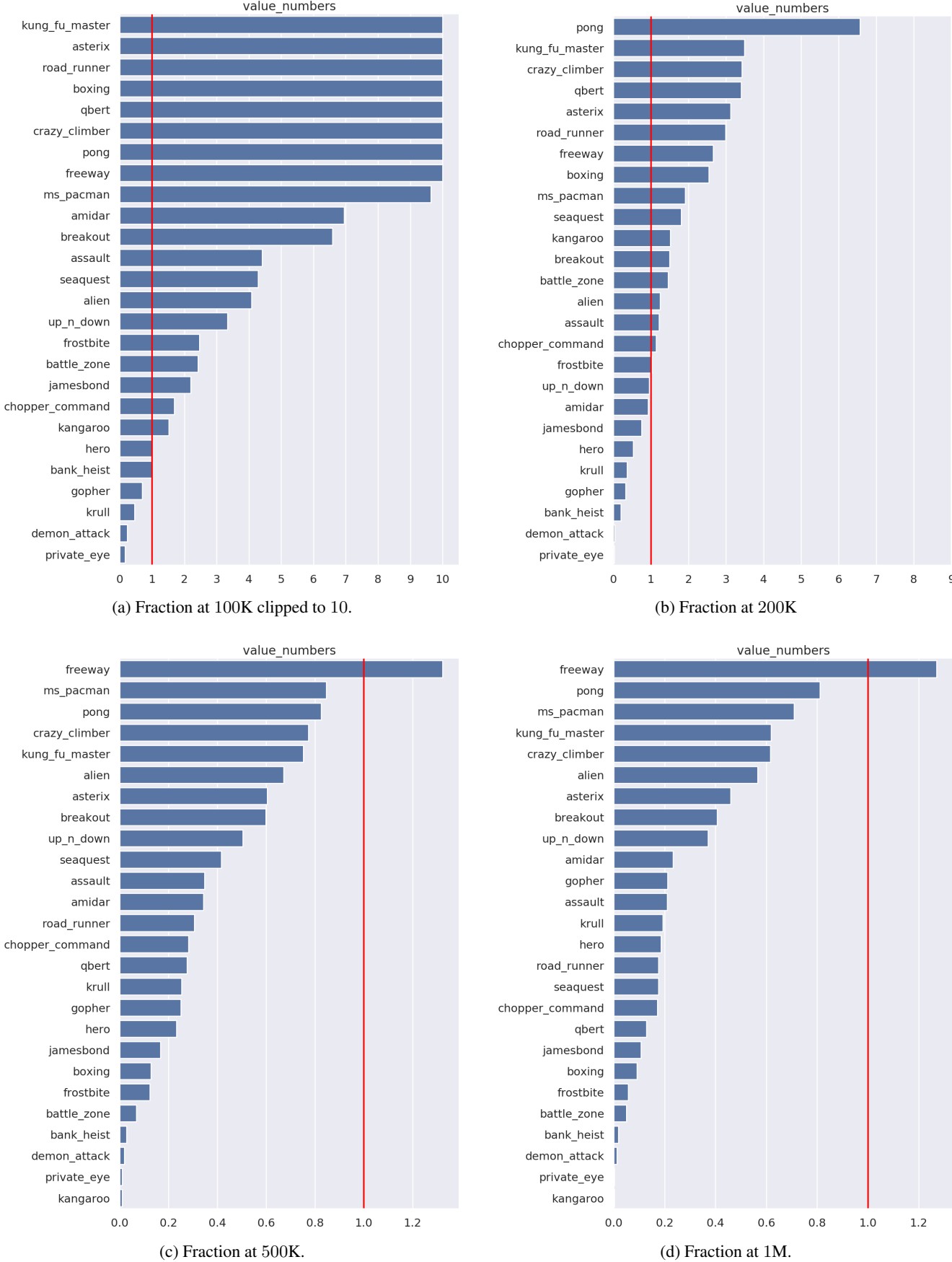

Figure 12: Fractions of the rainbow scores at given number of samples. These were calculate with the formula $(SimPLe\_score - random\_score)/(rainbow\_score - random\_score)$; if denominator is smaller than 0, both nominator and denominator are increased by 1.

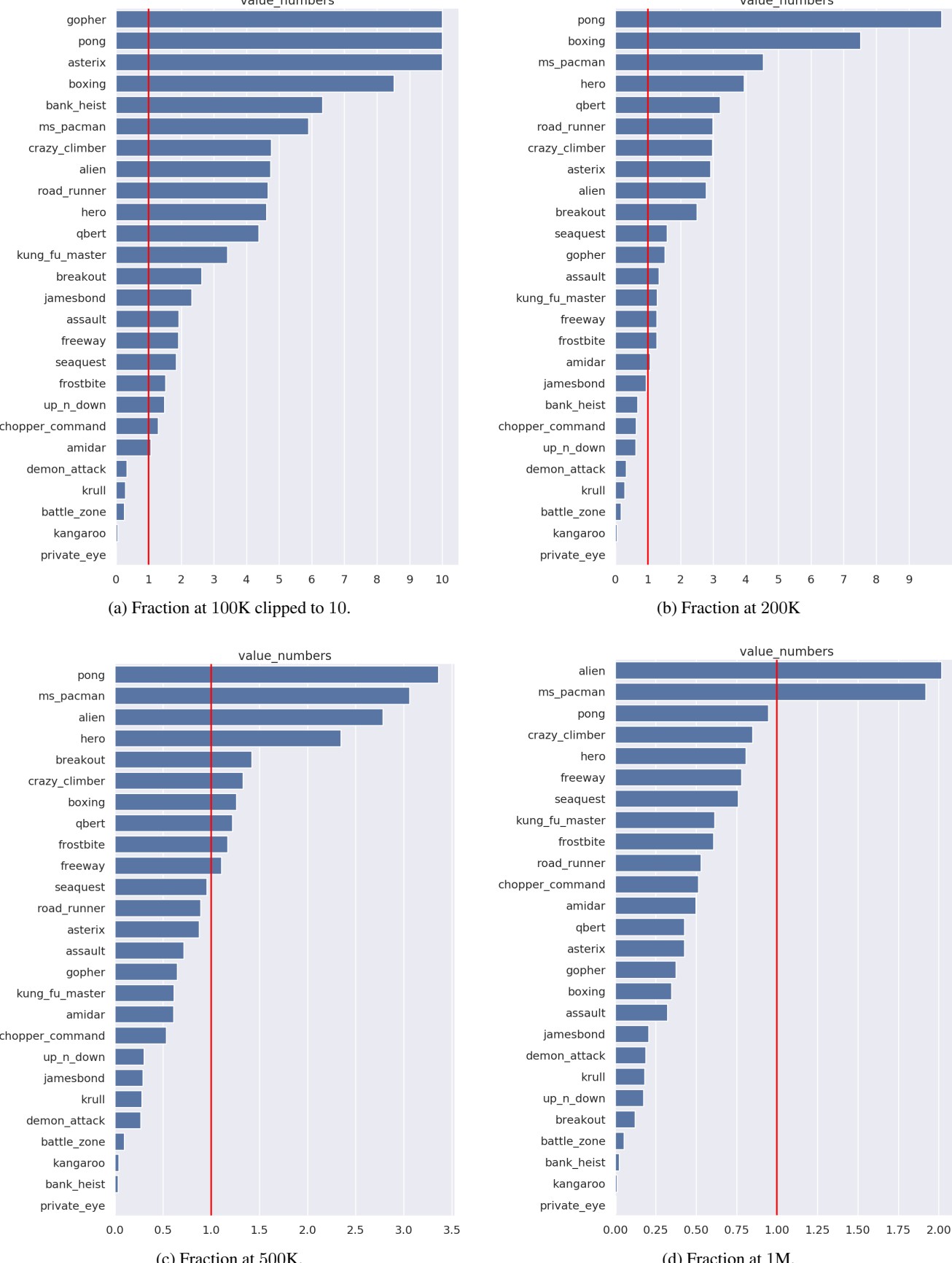

Figure 13: Fractions of the ppo scores at given number of samples. These were calculate with the formula $(SimPLe\_score - random\_score)/(ppo\_score - random\_score)$; if denominator is smaller than 0, both nominator and denominator are increased by 1.

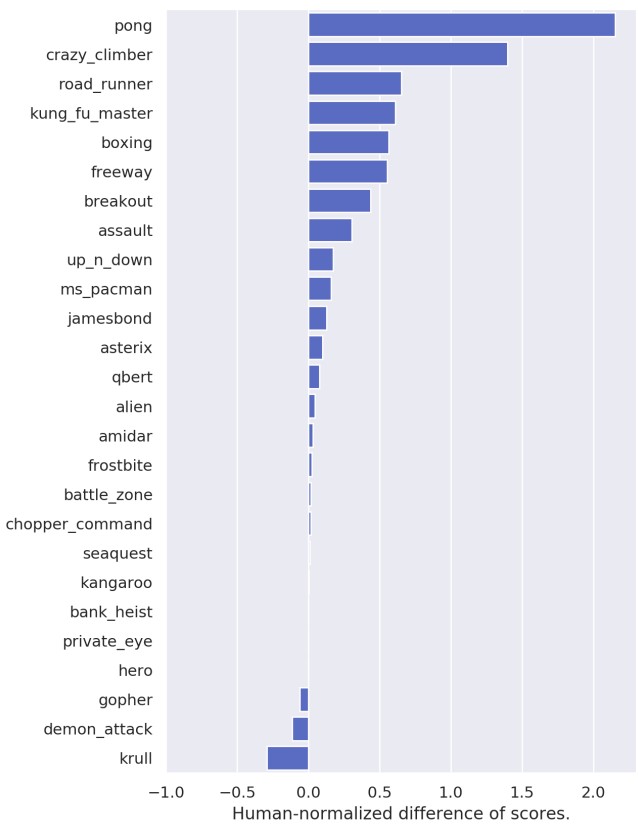

(a) SimPLe compared to Rainbow at 100K.

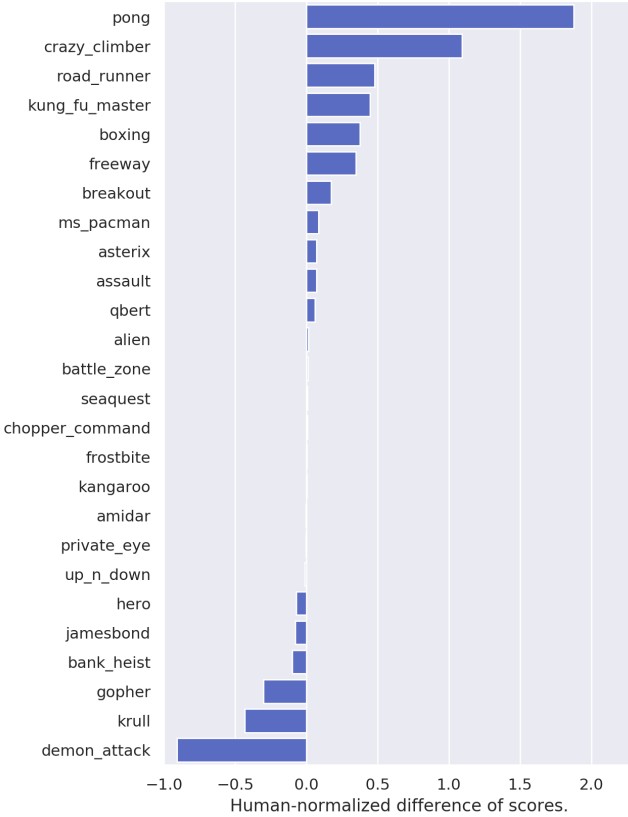

(b) SimPLe compared to Rainbow at 200K

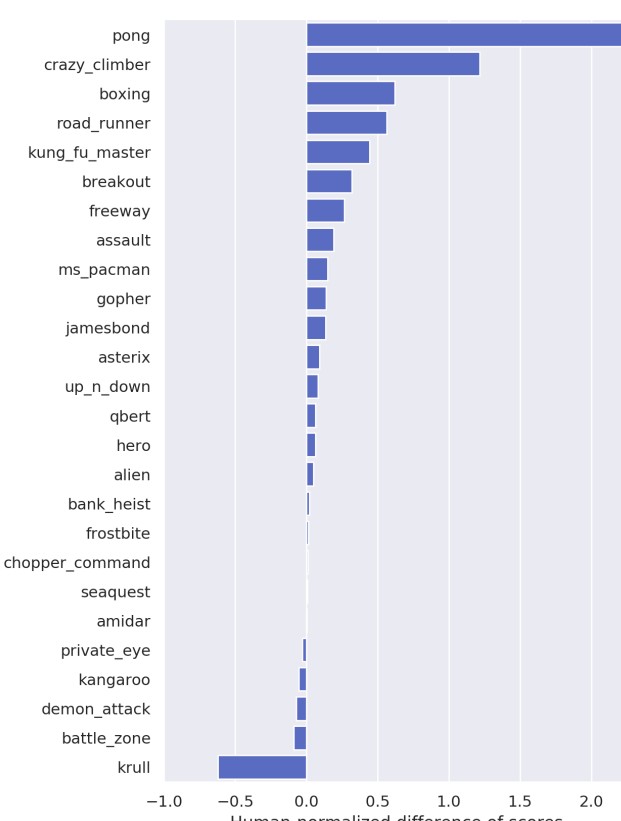

(c) SimPLe compared to PPO at 100K.

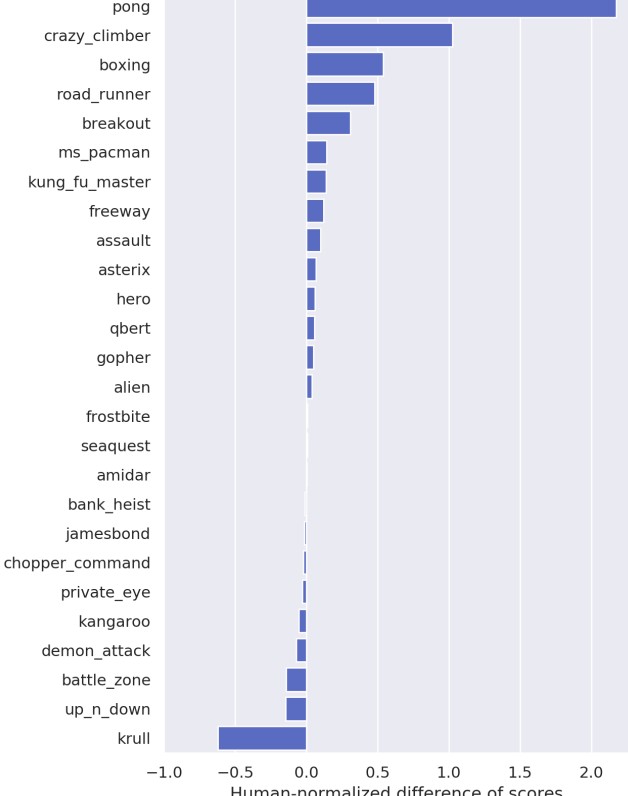

(d) SimPLe compared to PPO at 200K.

Figure 14: *Comparison of scores from Simple against Rainbow and PPO at different numbers of interactions. The following formula is used: $(SimPLe\_score@100K - baseline\_score)/human\_score$. Points are normalized by average human score in order to be presentable in one graph.*

