# OpenReview forum: "Model Based Reinforcement Learning for Atari"
_ICLR.cc/2020/Conference — Accept (Spotlight)_

### Official Review · AnonReviewer2 · 2019-10-17
**Official Blind Review #2**

**Rating:** 6

**Review:**

Summary

This paper proposes a model-based reinforcement learning algorithm suitable for high-dimensional visual environments like Atari. The algorithmic loop is conceptually simple and comprises 1) collecting real data with the current policy 2) updating an environment model with old and newly acquired data and 3) updating the policy "virtually" inside of the environment model using PPO. The approach is evaluated on 26 games from the Atari benchmark and compared against the model-free baselines Rainbow DQN and PPO. The newly proposed model-based method clearly outperforms both model-free baselines in low training regimes (100,000 steps). Further ablation studies are provided, e.g. similar results are obtained in a more stochastic setting of ALE with sticky actions.

Quality

This paper has a strong applied focus and needs to be judged based on its experiments. The quality of those are high. The method is evaluated on a suite of 26 games, compared to strong model-free baselines and results are averaged over 5 seeds. One concern I have is that the method is only evaluated in low training regimes. While I do understand that increasing the training horizon is computationally demanding, results in the appendix (Figure 11a) indicate that the proposed model-based method has worse asymptotic performance compared to the model-free baselines. After 500,000 training steps the effect of sample efficiency vanishes and the final performance results are far away from the final performance results of the model-free baselines after 50,000,000 training steps. Also, a plot similar to Figure 11a) from the appendix for Rainbow DQN would be good (but I do understand this might be difficult to obtain in the course of the review period should this require more experiments).

Clarity

The paper is clearly written and easy to follow. However, the authors could state in the main paper more clearly that their method excels in low training regimes and that the sample efficiency effect seems to vanish when increasing training iterations from 100,000 to 500,000 steps. In fact, Figure 11a) from the appendix should go into the main paper, and it should be also mentioned that there is a huge discrepancy between the maximum performance achieved by the proposed model-based method and the maximum performance achieved by the model-free baselines when training for 50,000,000 steps. Based on the experiments, it is not clear at all if the new method will eventually catch up with best model-free results from the literature when training time is increased, or stall in low-performance regimes indefinitely.

Originality

The originality of this paper is not very high since the proposed algorithm and its components are not novel (there might be some minor novelty in the environment model architecture). However, this paper should not be judged based on its originality but based on its significance.

Significance

A working model-based RL algorithm for Atari is clearly a huge gap in the current literature and this paper takes an important step towards this direction. Demonstrating improved sample efficiency compared to strong model-free baselines in low training regimes is a significant result. The significance is however decreased by the fact that the paper does not answer the question how to obtain good asymptotic performance that matches (or comes close to) model-free state-of-the-art results. I therefore vote for weak accept at this stage.

Minor details

On a side note, there are two citations missing related to model-based RL in visual domains:
- S. Alaniz. Deep Reinforcement Learning with Model Learning and Monte Carlo Tree Search in Minecraft. In the 3rd Multidisciplinary Conference on Reinforcement Learning and Decision Making (RLDM), 2017.
- F. Leibfried and P. Vrancx. Model-Based Regularization for Deep Reinforcement Learning with Transcoder Networks. In NIPS Deep Reinforcement Learning Workshop, 2018.

Update

After the author response, my review remains the same. I think this paper is worthwhile publishing at ICLR.

**Experience Assessment:**

I have published in this field for several years.

**Review Assessment: Checking Correctness Of Derivations And Theory:**

I assessed the sensibility of the derivations and theory.

**Review Assessment: Checking Correctness Of Experiments:**

I assessed the sensibility of the experiments.

**Review Assessment: Thoroughness In Paper Reading:**

I read the paper at least twice and used my best judgement in assessing the paper.

---

> ### Author Response · Authors · 2019-11-13
> **Answer to AnonReviewer2**
>
> Thank you very much for the time taken to produce this high quality review. We appreciate the comments and have included them in the current version of the paper (in particular we included the mentioned references and made information about the asymptotic performance more visible in the main text, now in Section 6.2).
>
> The aim of this work was to develop model-based methods using visual predictions. As pointed by the reviewer this is a huge gap in the current state-of-the-art. We consider our work to be the first step on a longer path. In particular, we consciously focused on the case of 100k frames, mostly for practical reasons of being computationally more feasible. On top of that, we provided results for other numbers of frames but without much tuning, which we believe would improve the performance. Having said that, most likely at the moment our method would not provide results matching state-of-the-art model free approaches. There are at least two reasons, one of a practical nature is that it is hard to compete with years of the model-free research in one step. The second is more fundamental, it is generally observed that model-based methods rarely compare with model-free ones as to the asymptotic performance [1].
>
> Other comments:
> * “should be also mentioned that there is a huge discrepancy between the maximum performance achieved by the proposed model-based method and the maximum performance achieved by the model-free baselines when training for 50,000,000 steps”
> In the summary of the submitted version of the paper, it was already stated that “the final scores are on the whole lower than the best state-of-the-art model-free methods”. As mentioned above we have now added Sec. 6.2. discussing performance with more frames in the main text to make it more evident.
>
> [1] Tingwu Wang et al., Benchmarking Model-Based Reinforcement Learning, https://arxiv.org/abs/1907.02057

---

### Official Review · AnonReviewer3 · 2019-10-22
**Official Blind Review #3**

**Rating:** 8

**Review:**

This paper covers the author’s approach to learning a model of a game, which can then be used to train a reinforcement learning agent. The major benefit of the approach is that instead of requiring millions of training steps in the game, the model can be constructed with only 1-2 hours of footage, and then train in the simulated game for millions of training steps.

This is a well-written paper, and the results are very impressive. The approach builds upon prior work with the same general thrust, but broadly makes clear that it stands above these existing approaches. However, I would have appreciated some clarity in the comparison made to the work of Ha and Schmidhuber (2018). It is unclear if the difference given is just because of the environments employed by Ha and Schmidhuber or if the authors see the approach presented in this paper as fundamentally different or improved in some way.

My one major technical concern comes down to how this work is framed and what that implies about appropriate baselines. The authors state clearly that the benefit of this work is that it can learn a sufficient model of a game in only 1-2 hours of gameplay footage. As I said above that is very impressive. However, the agents then requires 15.2 million interactions in this environment to learn to play the game. I would have appreciated some clarity then in the computational resource cost in this approach as opposed to just training say Rainbow in the actual game environments with 15.2 million interactions. It’s also not clear if optimizing Rainbow’s performance on 1M steps is a fair comparison. Ideally I would have liked to have seen some variance in the amount of time Rainbow was trained for compared to the associated computational costs. Clarity on this especially in sections like 6.1 would help readers better grasp the tradeoffs of the approach.

The authors could have also included reference to non-DNN work on learning forward/engine/world models that were then used to play or reason about the game. For example Guzdial and Riedl’s 2017 “Game Engine Learning from Gameplay Video” on Super Mario Bros. or Ersen and Sariel’s 2015 “Learning behaviors of and interactions among objects through spatio–temporal reasoning” on a novel game.


**Experience Assessment:**

I have published one or two papers in this area.

**Review Assessment: Checking Correctness Of Derivations And Theory:**

I did not assess the derivations or theory.

**Review Assessment: Checking Correctness Of Experiments:**

I assessed the sensibility of the experiments.

**Review Assessment: Thoroughness In Paper Reading:**

I read the paper at least twice and used my best judgement in assessing the paper.

---

> ### Author Response · Authors · 2019-11-13
> **Answer to AnonReviewer3**
>
> Thank you very much for the detailed review.
>
> We updated the paper to be more clear about the computational cost of our proposed method (conclusion and Appendix C).  In short, the computational resources required are higher than running a model-free algorithm (e.g. Rainbow) directly on the ALE environment. We aimed at developing a working model-based algorithm for the well-studied Atari domain. We ignored computational cost, though clearly it is to be addressed in the future. We also think that our method has additional benefits in environments where collecting real world experience is expensive or dangerous, such as robotics or autonomous driving.
>
> We would like to clarify the differences between our work and Ha and Schmidhuber (2018). Their work is clearly important and presents a similar direction of using a world model for RL.
> The first difference is in the architecture of world models. We have introduced a novel architecture with a stochastic discrete latent variable, inspired by both VAE and pixel-RNNs; based on our extensive experiments the architecture was crucial for the ALE domain (please refer to the ablation section of our paper).
> Moreover, we evaluated our approach on a broad set of Atari games coming from a much studied benchmark. These environments present a wide range of difficulties. In some cases, our method worked for games which require non-trivial exploration (as for example Freeway) and thus are particularly challenging for model-based methods (e.g. require the model to be consistent over an extended number of steps). Ha and Schmidhuber are training the world model only based on trajectories from random agents. In the case of ALE, we have verified that this is not enough and that better results are achieved when we repeat the loop of collecting experience, training the world model and training the agent multiple times (see. Figure 6(c) in the Appendix).
>
> Thank you also for providing the interesting references to work not using deep learning. We have included them in the updated version of the paper.

---

### Official Review · AnonReviewer1 · 2019-10-23
**Official Blind Review #1**

**Rating:** 6

**Review:**

The paper addresses sample-efficient learning (~2 hours of gameplay equivalent) for Atari (ALE) games. Building on the idea of training in a learned world model and the use of a u-net next-frame predictor, the approach is claimed to yield almost comparable performance to other models with only a fraction of the true-environment experience.

Sample efficiency is a major concern for DRL, particularly with an eye towards robotics and other physical domains. Although the approach is rather specific to the shapes and qualities of data in the ALE setting, the work is motivated at a high level, and the specific techniques for predicting the next frame explained in the past are explained.

This reviewer moves for a weak accept on account that the paper is well written (with quite thorough experiments explaining improvements in sample efficiency and possible limits in final task performance) but specifically targets ALE where execution is so cheap. The total number of PPO updates made in the new approach is not much reduced from before even if the number of trajectories evaluated in the true environment is very much reduced. On the problem of how much RL itself is sample efficient, not much progress is made.

Question:
- What is the impact on total wall-clock training time when using this approach? Given that the technique is centered on ALE, the characteristics of ALE compared to the learned world model are relevant (ALE executes very quickly and easily parallelizes whereas the learned world model presumably only runs where you have a GPU).
- Can this approach be stacked to benefit from training in a lighter-weight approximate model (env'') of the world model (env')?

**Experience Assessment:**

I have published one or two papers in this area.

**Review Assessment: Checking Correctness Of Derivations And Theory:**

N/A

**Review Assessment: Checking Correctness Of Experiments:**

I assessed the sensibility of the experiments.

**Review Assessment: Thoroughness In Paper Reading:**

I read the paper at least twice and used my best judgement in assessing the paper.

---

> ### Author Response · Authors · 2019-11-13
> **Answer to AnonReviewer1**
>
> We thank the reviewer for the valuable and detailed review.
>
> As the reviewer mentioned, sample efficiency is most important when collecting samples is hard including the application interacting with the physical world. And we also agree with the reviewer that ALE is not a good example of such an environment since collecting new trajectories in ALE is quite cheap. However, we chose ALE to demonstrate the capability of our proposed method in a setting with: 1) relatively complex high dimensional observation space and 2) task variety. We believe this setting demonstrates the generality of the method which potentially can be employed in the real world, where collecting samples is expensive, as well.
>
> > - What is the impact on total wall-clock training time when using this approach? Given that the technique is centered on ALE, the characteristics of ALE compared to the learned world model are relevant (ALE executes very quickly and easily parallelizes whereas the learned world model presumably only runs where you have a GPU).
>
> The wall-clock training time of SimPLe is increased over standard model-free training. We’ve updated the article with clear information about this in conclusions and Appendix C.
> Our aim was to develop a working model-based RL algorithm of general applicability and decrease sample efficiency in the low-samples regime. Having said that, we chose the Atari domain as a well-established testing ground and consciously ignored the computation issues. In fact the wall-time of our algorithm is higher than using standard-model free. We feel it is the price to be paid for the above mentioned improvements, though admittedly making models faster (light-weight) is a tempting research area!
> The world model, in theory, doesn’t need GPU to be trained and evaluated, but in practice, the runtime would increase even further if run without an accelerator.
>
> > - Can this approach be stacked to benefit from training in a lighter-weight approximate model (env'') of the world model (env')?
>
> This is an interesting idea, which we have not explored yet. However, we have observed that simplifications of our world model (e.g. not including stochastic discrete latent) produce significantly weaker predictions and agents trained inside of these world models don’t transfer well to the real environment.

---

### Public Comment · ~Kacper_Piotr_Kielak1 · 2019-10-23
**What's your opinion on recent MFRL results in the low-data regime?**

Thank you for submitting your study. I really enjoyed reading about your SOTA world model architecture, especially that I was experimenting with MBRL at the time. It would be interesting to see how it impacts the performance of other existing algorithms that employ learnable world models.

I'm curious about your thoughts on [1] and [2] though. Both papers conclude that utilizing the model in a way outlined by SimPLe does not provide an improvement over replay-based methods. Both of them employ quite simplistic tuning techniques to achieve at least equal performance to SimPLe in the low-data regime.

[1] https://arxiv.org/abs/1906.05243
[2] https://openreview.net/forum?id=Bke9u1HFwB

---

> ### Author Response · Authors · 2019-11-02
> **Comparison with new model-free results and underlining our main contribution**
>
> Thank you for pointing us to these articles. They are very interesting pieces of work, and we are happy that our research has “partially inspired” that work [1]. In the final version of the paper, we will include stronger baseline results proposed in [2].
>
> Moreover, we would like to point out that:
> 1. In our opinion, even with the model-free results presented in [1] and [2] our method can still be considered state-of-the-art with regards to the sample complexity on Atari. [2] clearly states that out of the 26 games tried there is a tie between two compared methods (SimPLe is better in 13, so is OTRainbow). [1] claims that their optimized version of Rainbow beats SimPLe at 17 out of 26 games, but this is because they are using an outdated version of our result. With the best results presented in this paper (coming from SD Long model), we are again on pair with [1]. Detailed comparison using our current results can be found:
> https://docs.google.com/spreadsheets/d/1uf5C79LeaDZfOwt_4Pm2R1dOXPPTTjM3ijkM95nc3H8/edit?usp=sharing
> Please note that the only difference between our older and newer (better) results is the increased training time of the world model, which clearly shows that with further hyperparameter optimization SimPLe could score even better.
>
> 2. It is no secret that RL methods behave quite differently with various random seeds and hyper parameters. We searched for optimal model-free baseline much more than we searched for optimal hyperparameter set for SimPLe. We believe both can be improved with a more rigorous search. We also would like to mention that the results in [2] are similar to the ones we reported in our paper. SimPLe has a hard time learning in environments that we consider hard and are discussed, in detail, in our analysis such as bankheist, games based on exploration: hero, private eye, james bond.
>
> 3. Please note that one of the main focuses of our paper and our proposed method is to demonstrate the possibility of using model-based reinforcement learning for ATARI where observation space is huge and tasks vary significantly. We illustrated how such models can be scaled to this kind of problem, something that has not been shown before.
>
>
> [1] https://arxiv.org/abs/1906.05243
> [2] https://openreview.net/forum?id=Bke9u1HFwB

---

### Decision · Program_Chairs · 2019-12-19

**Decision:**

Accept (Spotlight)

**Comment:**

This paper presents a model-based RL approach to Atari games based on video prediction. The architecture performs remarkably well with a limited amount of interactions.  This is a very significant result on a question that engages many in the research community.

Reviewers all agree that the paper is good and should be published. There is some disagreement about the novelty of it. However, as one reviewer states, the significance of the results is more important than the novelty. Many conference attendees would like to hear about it.

Based on this, I think the paper can be accepted for oral presentation.